# Translation regulatory factor BZW1 regulates preimplantation embryo development and compaction by restricting global non-AUG Initiation

Jue Zhang[1,2,3,4,5], Shuai-Bo Pi [2,5], Nan Zhang[3], Jing Guo[1,3], Wei Zheng[1,3], Lizhi Leng[1,3], Ge Lin [1,3] ✉ & Heng-Yu Fan [2] ✉

Protein synthesis is an essential step in gene expression during the development of mammalian preimplantation embryos. This is a complex and highly regulated process. The accuracy of the translation initiation codon is important in various gene expression programs. However, the mechanisms that regulate AUG and non-AUG codon initiation in early embryos remain poorly understood. BZW1 is a key factor in determining the mRNA translation start codon. Here, we show that BZW1 is essential for early embryonic development in mice. *Bzw1*-knockdown embryos fail to undergo compaction, and show decreased blastocyst formation rates. We also observe defects in the differentiation capacity and implantation potential after *Bzw1* interference. Further investigation revealed that *Bzw1* knockdown causes the levels of translation initiation with CUG as the start codon to increase. The decline in BZW1 levels result in a decrease in protein synthesis in preimplantation embryos, whereas the total mRNA levels are not altered. Therefore, we concluded that BZW1 contributes to protein synthesis during early embryonic development by restricting non-AUG translational initiation.

During the preimplantation stage of mammalian development, compaction is the first critical morphological event that occurs at the 8-cell stage. It involves the smoothening of the embryo's surface, which is associated with an increase in intercellular adhesion mediated by E-cadherin (E-cad)[1,2]. The establishment of cell polarity occurs concomitantly with compaction, and it is initiated de novo at the 8-cell stage[3,4]. The blastomeres located inside of the morula give rise to the inner cell mass (ICM), whereas the outer blastomeres exclusively differentiate into the trophectoderm (TE)[5]. During this process, proteins, that are either inherited from oocytes or are newly formed after zygotic gene activation (ZGA) execute most biological programs to ensure normal embryonic development[6–8]. Therefore, determining the factors that are crucial for regulation of protein synthesis will provide perspective for improving early embryo quality.

Many translational regulation proteins have been shown to contribute to early embryo compaction and cell fate determination. For example, the nucleolar protein DDB1 and CUL4 associated factor 13 (DCAF13) has been suggested to play an important role in ribosomal RNA processing. Studies have shown that *Dcaf13*[-/-] embryos fail to undergo compaction and exhibt decreased protein levels and cell differentiation[9,10]. Eukaryotic translation initiation factor 4E (eIF4E) mediates cap-dependent translation. Its absence has been shown to

[1]Clinical Research Center for Reproduction and Genetics in Hunan Province, Reproductive and Genetic Hospital of CITIC-XIANGYA, 410078 Changsha, China. [2]Life Sciences Institute, Zhejiang University, 310058 Hangzhou, China. [3]NHC Key Laboratory of Human Stem and Reproductive Engineering, School of Basic Medical Science, Central South University, 410078 Changsha, China. [4]College of Life Science, Hunan Normal University, 410006 Changsha, China. [5]These authors contributed equally: Jue Zhang, Shuai-Bo Pi. ✉e-mail: linggf@hotmail.com; hyfan@zju.edu.cn

result in peri-implantation embryonic lethality due to the failure of normal epiblast formation[11]. According to the protein expression profiles in every early embryo stage of mice, proteins involved in translational initiation are continuously upregulated during pre-implantation development[12]. This suggests that regulatory mechanisms of translational initiation may play an essential role in early mammalian embryos. However, few relative molecular studies have been conducted on this topic.

Basic leucine zipper and W2 domains 1 (BZW1) is a zygotic gene that is highly expressed in both oocytes and preimplantation embryos. This protein was first isolated from HeLa S3 cells by DNA affinity chromatography, which is thought to enhance H4 gene transcription[13]. Subsequent studies have established that BZW1 is a biomarker and tumors therapeutic target[14–16]. However, the physiological function of BZW1 in early mouse embryonic development has not yet been clarified. Because of BZW1 contains a W2 domain that is usually found at the C-terminus of several translation initiation factors, such as eIF4G and eIF5. Previous studies have suggested that BZW1, as well as its paralog BZW2, is an eIF5-mimic protein that acts as a translational rheostat[17]. eIF5[18] and BZW1/2[19] can also autoregulate their own translation in HEK293T cells. In this study, we hypothesized that BZW1, as a translation initiation regulation factor, plays an important role in preimplantation embryo protein synthesis.

Translation initiation begins with the formation of a ternary complex (TC) of methionyl initiator tRNA (Met-tRNA$_i$), eukaryotic initiation factor 2 (eIF2) and GTP, which assemble with the 40 S ribosomal subunit and eukaryotic translation initiation factors (eIF1, 1 A, 3, and 5) to form the 43 S pre-initiation complex (PIC)[20]. When bound at the 5′ end of the mRNA, the PIC scans to locate the start codon, while eIF5 is a crucial component in the promotion of start codon selection[21]. Traditionally, initiation of translation of protein-coding sequences in the start codon selection (ORFs) begins at the universal AUG start codon. However, ribosome profiling studies have shown that initiation at non-AUG codons (e.g., CUG, GUG, and UUG) prevalently occurs in mammalian cells, not as an error, but as a mechanism to regulate protein translation[20,22]. It has been reported that elevated eIF5 levels can enhance the initiation with CUG or GUG codons[21,23]. In contrast, in vitro experiments have shown that the addition of eIF1 and eIF1A can result in fewer initiations at CUG codons and more initiations at AUG codons[23].

In this work, we investigated the role of BZW1 in preimplantation embryonic development and compaction, in relation to the accuracy of start codon selection. We examined the expression levels of BZW1 protein during embryo development and knocked down Bzw1/2 from zygotes by RNA interference (RNAi) to investigate their function in preimplantation embryonic development. Thus, we attempted to elucidate the molecular mechanisms involved in this process.

## Results

### BZW1 is distributed in the cytoplasm and highly expressed in murine pre-implanted embryos

Analysis of the expression profiles of Bzw1/2 in early mouse embryos using quantitative reverse transcription-polymerase chain reaction (RT-PCR) revealed that Bzw1/2 mRNA expression increased from the 2-cell stage and reached a peak at 4-cell stage (Fig. 1a). We also found that Bzw1 mRNA expression was higher than that of Bzw2 from the zygote to the 4-cell stage. Western blot results showed that the BZW1 protein was translated as early as the zygote stage, increased rapidly at the 8-cell stage, and then continued to accumulate in blastocysts (Fig. 1b). Immunofluorescent staining also revealed that BZW1 was expressed during early embryonic development, and that it was located in the cytoplasm of blastomeres from 2-cell to blastocyst stage (Fig. 1c, d). Overexpression of mCherry-BZW1 in embryos confirmed the specificity of the BZW1 antibody (Fig. S1a). We also examined the subcellular localization of BZW1 in HeLa cells and observed that the

overexpressed FLAG-BZW1 was also localized in the cytoplasm (Fig. S1b).

Bzw1 and its homologous gene Bzw2 are conserved across many organisms from yeast to mammals. Mouse BZW1 and BZW2 share 73% sequence identities, and both contain two HEAT domains, a MA3 domain and a W2 domain. The expression patterns of BZW1/2 suggest that they may play vital roles in preimplantation embryos. Based on the mRNA levels, the function of BZW1 was probably predominant during this process.

### BZW1 is crucial for early embryonic development and compaction in mice

We employed an RNAi approach to study the function of BZW1 during preimplantation development, by injecting Bzw1-targeting small interfering RNAs (siBzw1s) into the cytoplasm of wild-type (WT) zygotes. Western blotting showed that the BZW1 protein levels decreased at the 8-cell stages in siBzw1s-injected embryos (Fig. 2a). RT-PCR revealed that Bzw1 mRNA expression was significantly repressed at the 2-cell stages after siBzw1-injection but Bzw2 levels was not affected (Fig. 2b). The embryos were cultured for 3.5 days in vitro. We observed that the embryo development was repressed at every stage, and the rate of compaction decreased markedly after Bzw1 interference (Fig. 2c, d). Bzw2 gene, which is also expressed during pre-implantation embryos development (Fig. 1a), shares high homology with Bzw1. We verified the effects of siBzw2 on embryonic development (Fig. 2c, d). RT-PCR showed that Bzw2 mRNA level were significantly reduced at the 2-cell stages after siBzw2 injection, but Bzw1 level was slightly increased (Fig. 2e). We found that Bzw2 decrease had little effect on embryo compaction but decreased blastocyst developmental rate significantly (Fig. 2d, f, g). Bzw1 and Bzw2 siRNAs were also co-injected into zygotes. The rate of compaction declined in the double-knockdown embryos, and these embryos failed to develop to blastocyst (Fig. 2c, d, f, g). To further clarify the relationship between the function of BZW1 and BZW2, we showed that BZW1 could partly rescue the development rate of siBzw2 embryos by expressing BZW1 in Bzw2 knockdown embryos (Fig. S2a–d); BZW2, in turn, could also rescue the siBzw1 embryo compaction and development (Fig. S2a–e). Therefore, a certain level of functional compensation between existed between BZW1 and BZW2.

Notably, most Bzw1-knockdown embryos failed to undergo compaction (Fig. 2c, d). Early embryo compaction occurs at the 8- to 16-cell stage. During this process, cells flatten their membranes against each other, leading to cell–cell contact elongation and the sealing of adjacent blastomeres. To quantify the degree of compaction in Bzw1-knockdown embryos, we removed the zona pellucida of embryos using acidic M2 medium (pH 2.0) at the 8- to 16-cell stages. We found that most of the control blastomeres were not segregated. However, more than half of the blastomeres in Bzw1-knockdown embryos were segregated to various degrees (Fig. 3a, b).

When compaction occurs, blastomeres extend long E-cad-dependent filopodia on to neighboring cells to control cell shape changes and cell–cell adhesion[1]. It is known that treatment of cells with 1% Triton X-100 can dissolve away the membranes and soluble components, leaving intact cytoskeletal residue[24]. We treated embryos with Triton X-100 then examined the localization of E-cadherin in the cytoskeleton to reflect filopodia formation. Immunofluorescence staining showed that the total E-cad protein level was unchanged (Fig. 3c, d) but the level of E-cad remaining in the cytoskeleton was lower in the Bzw1-knockdown embryo than in the negative control (Fig. 3c, e).

Embryo compaction coincides with the acquisition of cell polarity. This leads to the asymmetric distribution of cellular components, which is critical for cell fate specification. Ezrin is a member of the ERM protein family, that serves as a crosslinker between F-actin and the plasma membrane. The crosslinker function of Ezrin requires the phosphorylation of T567[25,26]. The pEzrin$^{T567}$ was detected at the early 8-cell

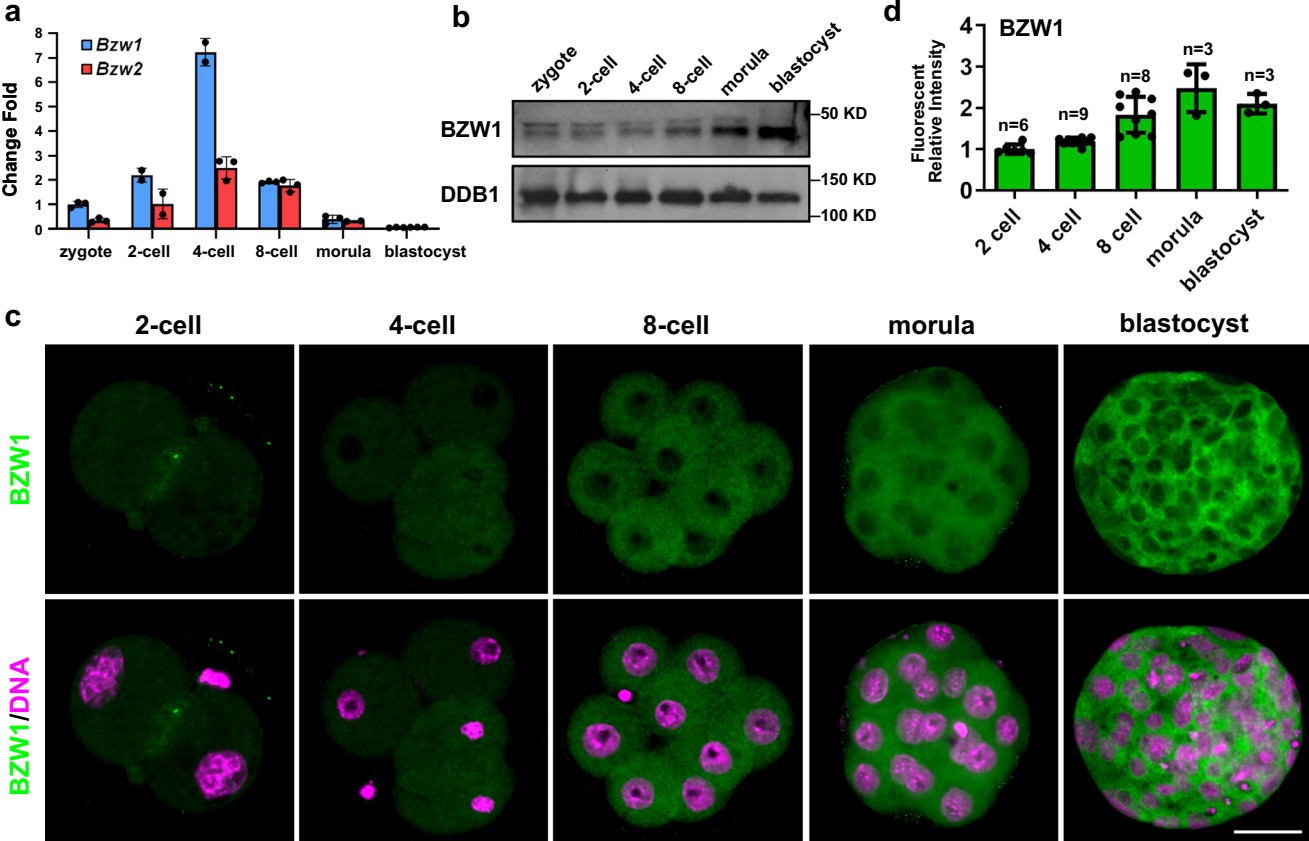

**Fig. 1 | BZW1 is a cytoplasmic protein that is continuously expressed during embryonic development. a** Absolute quantitative RT-PCR results for *Bzw1* and *Bzw2* expression in mouse preimplantation embryos. The expression levels at each developmental stage were normalized to *Gapdh*, which served as an internal control. The relative mRNA level of *Bzw1* at the zygote stage was defined as 1.0. Data are presented as mean values. Error bars indicate the standard deviation (SD) (five embryos were collected at each time point). **b** Protein levels of BZW1 in mouse preimplantation embryos were determined by western blot analysis. DDB1 was blotted as a loading control. Total protein samples from 100 embryos were loaded in each lane. The sizes (kDa) of the protein markers are indicated on the right-hand side. The experiment was repeated two times with similar results. **c** Confocal microscopic images of BZW1 (green) immunofluorescence in preimplantation embryos from 2-cell stage to the blastocystic stage. DNA was counterstained with 4′, 6-diamidino-2-phenylindole (DAPI, magenta). Scale bar, 25 μm. **d** Quantification of BZW1 signal intensity in **c**. *n* indicates the number of embryos were analyzed. The relative immunofluorescence level of *Bzw1* at the zygote stage was defined as 1.0. Data are presented as mean values. Error bars indicate SD. All source data are provided as a Source Data file.

embryo stage, which mediates the formation of F-actin at the apical surface, and promotes embryo compaction and polarization[25,26]. Ezrin is distributed on the cell-contact-free surface at the late eight-cell stage, then is apically distributed in compacted embryos[3]. We observed the level and localization of pEzrin[T567] to reflect the degree of compaction and polarity. Considering that the time span of compaction is relatively short and embryo development is difficult to synchronize completely, we co-injected mCherry mRNAs and si*Bzw1*s into one blastomere at the 2-cell stage embryos, to reduce *Bzw1* expression in half of the embryos. The endogenous pEzrin[T567] were stained when compaction occurs in the control embryos. During apical protein polarization, pEzrin concentrated at the center of the cell-contact-free surface to form an apical patch (Fig. 3f, g-I). We found that pEzrin signal in the half of the embryos that *Bzw1* knockdown was uniformly distributed on the cell-contact-free surface (Fig. 3f, g-II) and in the other half embryo became restricted to the apical region (Fig. 3f, g-III). The pEzrin signal intensity in half *Bzw1*-knockdown embryos was lower than that in control embryos (Fig. 3h). These results revealed that declining BZW1 levels impaired embryo polarity and may affect cell differentiation.

## BZW1 regulates differentiation capacity and implantation potential of embryos

We observed that some *Bzw1*-knockdown embryos were still able to reach the blastocyst stage morphologically. As the degree of embryo

polarity in the *Bzw1*-knockdown morula stage decreased (Fig. 3f), we suspected that the differentiation of preimplantation embryos could be regulated by BZW1. Immunofluorescence staining revealed the levels and localization of OCT4, NANOG, and CDX2 proteins, all of which are essential for the differentiation of ICM and TE[27,28]. The results suggest that OCT4 and NANOG protein levels decreased in *Bzw1*-knockdown embryos with irregular localization during the 8-cell to blastocyst stages, whereas CDX2 was unaffected (Fig. 4a). We proposed that decreasing *Bzw1* suppresses zygotic genome activation (ZGA). Therefore, RT-PCR analyses of control and RNAi embryos were performed at the 4-cell and morula stages (Fig. 4b, c). Specifically, we examined the expression levels of transcription factors (*Nanog, Oct3/4, Klf5, Cdx2,* and *Sox2*), as lineage-specific genes that reflect the pluripotency[27,28] after *Bzw1* RNAi (Fig. 4c) at the morula stage. We found that the expression of ICM-dominant genes, including *Nanog, Oct3/4,* and *Klf5*, decreased significantly (Fig. 4c) in the RNAi morula, whereas the expression of *Sox2* and the TE-specific gene *Cdx2* was not affected. Other zygotic genes, including *Fgfr1, Fgfr2, Gjb5,* and *Dusp6*[12,29–31], which ideally should have been highly expressed from the 8-cell to the morula stage, showed decreased transcription levels at the morula stage (Fig. 4c). However, the mRNA levels of these genes did not exhibit any significant change at the 4-cell stage. This suggests that the decrease in BZW1 had no significant effect on transcription in the early stages after *Bzw1* knocked down. But as embryo development,

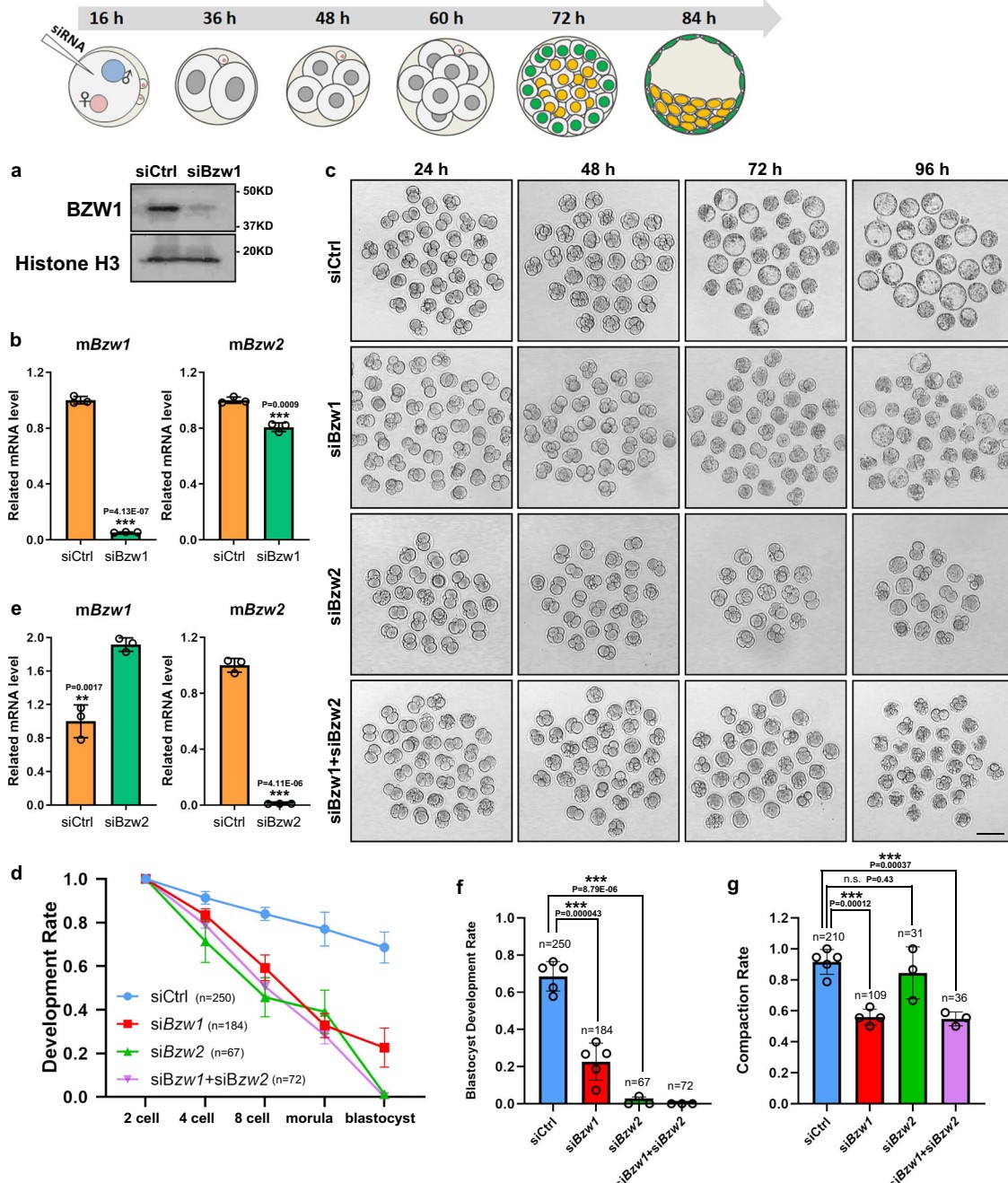

**Fig. 2 | BZW1 plays a key role in murine preimplantation embryonic development.** **a** The level of BZW1 in embryos obtained by western blotting at the 8-cell stage after microinjection of siControl or si*Bzw1* in zygotes, which also confirmed the specificity of the BZW1 antibody. Histone H3 was used as loading control. Total protein samples from 80 embryos were loaded in each lane. The experiment was repeated two times with similar results. **b** RT-PCR results showing the mRNA levels of *Bzw1* and *Bzw2* in embryo at 2-cell stage after microinjection of si*Bzw1* in zygotes. *Gapdh* served as the internal control. *n* = 5 embryos in every sample. The experiment was repeated three times. Data are presented as mean values. The relative mRNA level of *Bzw1* in control is defined as 1.0. ***P < 0.001, by two-tailed Student's *t*-tests. Error bars, S.D. **c** In vivo fertilized eggs were collected from oviducts, microinjected with siControl or the indicated siRNA, and then cultured for 3.5 days in vitro. DIC images of the embryos that reached the 4-cell, 8-cell, morula, and blastocyst stages after microinjection. Scale bar, 25 μm. **d** The percentage of embryos microinjected with siControl or indicated siRNA that reached

the 2-cell, 4-cell, 8-cell, morula, and blastocyst stages at 36, 48, 60, 72, and 84 h post-hCG injection was calculated and presented as the developmental rate. The experiment was repeated at least three times. Data are presented as mean values. Error bars indicate SD. **e** RT-PCR results showing the mRNA levels of *Bzw1* and *Bzw2* in embryo at 2-cell stage after microinjection of si*Bzw2* in zygotes. *n* = 5 embryos in every sample. *Gapdh* served as the internal control. Data are presented as mean values. The relative mRNA level of *Bzw1* in control is defined as 1.0. ***P < 0.001, by two-tailed Student's *t*-tests. Error bars, S.D. **f** Blastocyst developmental rates of embryos with microinjection of siRNA as in **d**. The experiment was repeated at least three times. *n* indicates the number of embryos were counted. Data are presented as mean values. ***P < 0.001, by two-tailed Student's *t*-tests. Error bars, S.D. **g** The compaction rates of embryos with microinjection of siRNA as in **d**. *n* indicates the number of 8-cell embryos were counted. Data are presented as mean values. n.s. not significant, ***P < 0.001, by two-tailed Student's *t*-tests. Error bars, S.D. All source data are provided as a Source Data file.

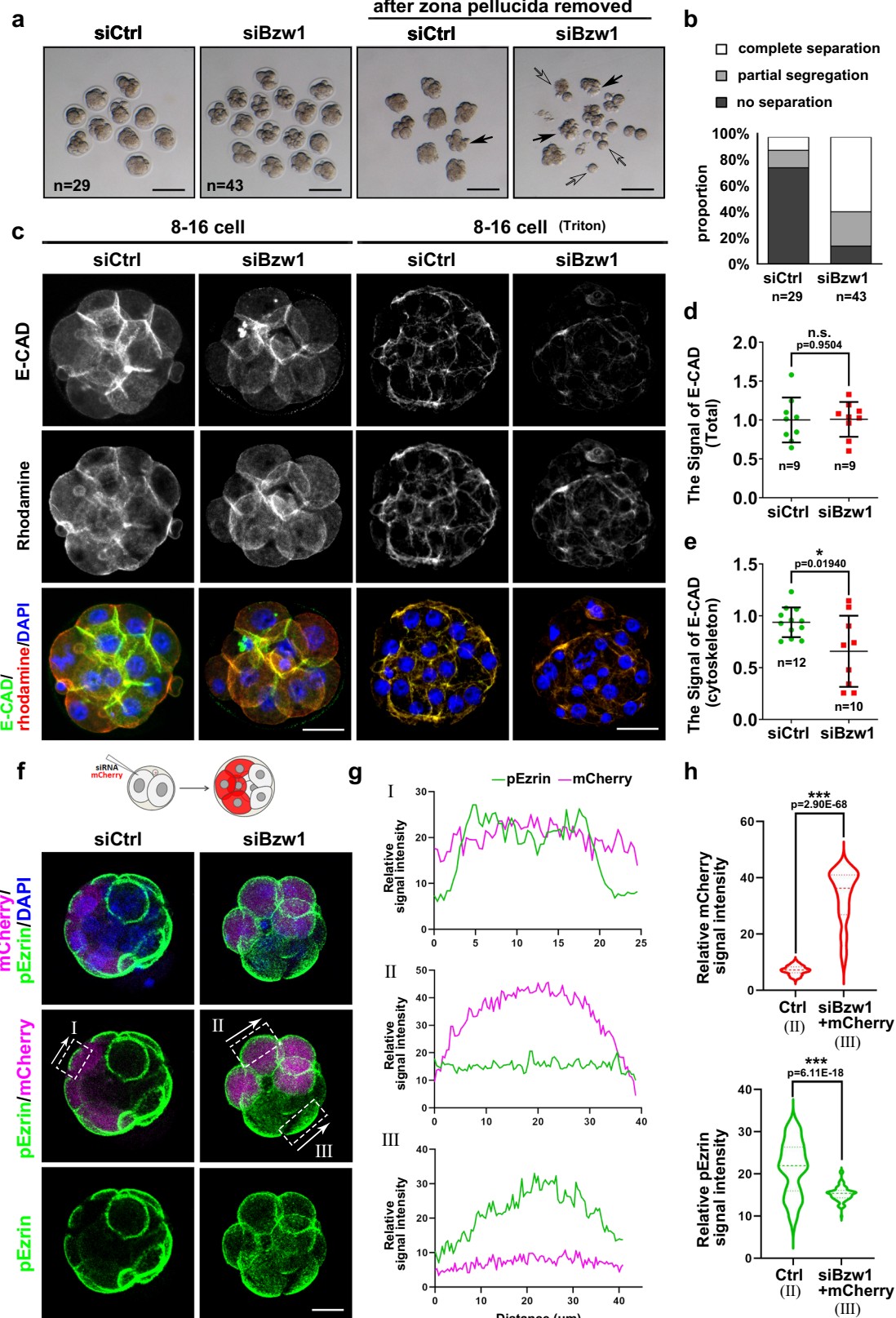

the embryos became abnormal and the expression of some ZGA genes was downregulated at the morula stage.

Based on the above results, we hypothesized that although a minority of *Bzw1*-knockdown embryos developed into blastocysts without obvious morphological defects (Fig. 2d), they may have suffered from developmental problems, especially the blastomeres in

ICM. Therefore, we further assessed these blastocysts using a 3-day outgrowth (OG) assay (Fig. S3a). Normally, blastocysts hatch out of the zona pellucida by 24 h, attach to the culture plate by 48 h and then form ICM colonies with surrounding trophoblast cells at 72 h. We found that 36.4% of the *Bzw1*-knockdown blastocysts could not hatch from the zona pellucida (Fig. 4d). Others attached to the culture plate,

**Fig. 3 | BZW1 is vital for preimplantation embryonic compaction.**
**a** Representative images showing the development of preimplantation embryos at the late 8-cell stage approximately 65 h post-hCG injection with or without *Bzw1* knockdown. The zona pellucida of the embryos was removed using an acidic M2 medium (pH 2.0). Digital image correlation (DIC) images of the embryos were obtained. Black arrows indicate partially segregated blastomeres in the embryos. Hollow arrows indicate blastomeres that were completely segregated after zona pellucida removal. Scale bar, 100 μm. **b** Proportions of blastomeres of siControl or si*Bzw1* embryos with different extents of separation after zona pellucida removal. **c** Confocal microscopic images of E-CAD (green) immunofluorescence preimplantation embryos at the 8-cell to 16-cell stage after siControl or si*Bzw1* microinjection in zygotes. Rhodamine-phalloidin (red) staining indicates actin cytoskeletal formation. The embryos were treated with 1% Triton X-100 (Sigma T8787) for 15 min before fixation to observe cytoskeleton protein (graph on the right). The DNA was counterstained with DAPI (blue). The greyscale pictures were shown for single channel. Scale bar, 25 μm. **d** Signal intensities of E-CAD from (c)

indicate the level of E-CAD in siControl and si*Bzw1* embryos fixed without Triton treatment. Data are presented as mean values. The relative Signal level of E-CAD in control is defined as 1.0. n.s. not significant, by two-tailed Student's *t*-tests. Error bars, SD. **e** Signal intensities of E-CAD from **c** indicate the level of E-CAD in siControl and si*Bzw1* embryos fixed after Triton treatment. Data are presented as mean values. The relative Signal level of E-CAD in control is defined as 1.0. *$P < 0.05$, two-tailed Student's *t*-test. Error bars, SD. **f** Confocal microscopic images of pEzrin[T567] (green) and mCherry (magenta) immunofluorescence in embryos that co-injected with mCherry mRNAs and si*Bzw1*s into one blastomere at the 2-cell stage, fixed when compaction occurred in the control embryos. The DNA was counterstained with DAPI (blue). Scale bar, 25 μm. White squares labeled I–III denote signal intensity analysis regions in **f**; Arrow direction corresponds to the *X*-axis in **f**. **g** Signal intensities of pEzrin[T567] and mCherry at the cell-contact-free surface in **f**. **h** The violin graph indicates the signal intensities of total level of pEzrin level at the cell-contact-free surface in **f**. ***$P < 0.001$, two-tailed Student's *t*-test. All source data are provided as a Source Data file.

but only 23.4% exhibited complete outgrowth (Fig. 4e). We also compared the proportion of blastocysts formed ICM colonies or trophoblast cells between control and si*Bzw1* groups. We found that approximately 77% of *Bzw1*-knockdown blastocysts that hatched out failed to form ICM colonies (Fig. 4f), which was significantly higher than that in the control group. But the proportions of trophoblast cell adhesion and development were not significantly different (Fig. 4g). The results also reflected the RT-PCR results that the expression of ICM-dominant genes decreased significantly.

We concluded that some *Bzw1*-knockdown embryos failed to compact and lost their apical-basal polarity, and the differentiation capacity and implantation potential of RNAi embryos reaching the blastocyst stage also declined.

## BZW1 ensures embryonic compaction and development by competing with eIF5 that decreases the accuracy of initiation codon selection

Previous studies have suggested that BZW1, as well as its paralog BZW2, is an eIF5-mimic protein that acts as a translational rheostat, increasing the accuracy of translation initiation by impeding eIF5-dependent translation initiation from non-AUG codons[17,19]. These studies found that the overexpression of eIF5 in HEK293T cells increased CUG, GUG, or UUG codon-initiated translation. First, we confirmed that eIF5 is expressed during early embryonic development according to the published data[32,33]. The transcription and translation levels of eIF5 were higher than those of BZW1/2 from zygote to morula stage (Fig. 5a, b). Then we tested whether improving eIF5-dependent translation initiation impairs early embryonic development. We overexpressed eIF5 in embryos by microinjecting green fluorescent protein (GFP)-eIF5 mRNA into zygotes (Fig. S4a). The results showed that overexpression of eIF5 decreased the rate of blastocyst development and compaction (Fig. 5c–f, S4b) and that E-CAD was localized in the cytoskeleton (Fig. 5g–i). These phenotypic features were similar to those observed in *Bzw1*-knockdown embryos (Fig. 2a, b). However, expression of some key cell fate commitment-related proteins and cell polarity in eIF5-overexpressed embryos were not affected (Fig. S5a–e).

We then determined whether BZW1 could rescue the developmentally defective phenotype caused by eIF5 overexpression and found that BZW1 and eIF5 mRNA co-injection improved the rate of blastocyst development, compared to eIF5 overexpression alone (Fig. 6a, e). Previous studies have suggested that BZW2 competes with eIF5 depending on the binding of its C-terminal domain (CTD) to the PIC through eIF2[34,35]. We conducted a competition assay by co-immunoprecipitation (co-IP) (Fig. 6b). The interaction between BZW1 and PIC was suppressed when eIF5 expression was increased. Our co-IP results revealed that BZW1 interacted with PIC through seven residues in its CTD as well (Fig. 5c, d). By comparing the rates of development

and compaction between BZW1-WT and BZW1-7A mutants co-expressed with eIF5, we found that BZW1 could rescue the defects caused by eIF5 overexpression to a certain extent but BZW1-7A mutants did not (Fig. 5e–g).

## BZW1 knockdown in preimplantation embryos increases non-AUG translation initiation and decreases translation efficiency

Global translation sequencing in mammalian cells has provided information regarding the position and density of ribosomes on mRNAs[36]. Overall, we found that 3066 genes overlapped between the transcriptomes expressed in mouse early embryo development (8990 RNA RPKM > 10 in at least one sample for each stage) and mRNA being identified as translation initiation site (TIS) positions by global translation initiation sequencing (GTI-Seq) in a mouse embryonic fibroblast (MEF) cell line (4147 genes) (Fig. S4a)[12]. According to the statistical results, we found that 1218 genes may contain at least one upstream TIS (uTIS) with non-AUG as the start codon in preimplantation embryo development. We also analyzed the proportion of non-AUG-initiated transcripts that were upregulated at every preimplantation embryo development stage (Fig. S4b). The stages from zygote to 2-cell had the highest proportion of transcripts containing uTIS with a non-AUG start codon.

We hypothesized that BZW1 would ensure the accuracy of initiation codon selection by decreasing non-AUG codon-initiated translation. We detected the phosphorylation of the Pol II CTD repeat (pRPS II) by immunofluorescence as an indication of Pol II activity in BZW1 RNAi embryos, suggesting that global transcription activity was not affected by *Bzw1* knockdown (Fig. S4c). RNA-sequencing (RNA-seq) results (Supplementary Data 1) also confirmed that most of the detected transcripts were unchanged in si*Bzw1* embryos (Fig. S6d), which also agreed with our RT-PCR results in the 4-cell stage (Fig. 4b). In addition, in conventional AUG initiation codons, CUG was the most prominent start codon among the non-AUG that started mRNA upregulated at every stage (Fig. S6e). Therefore, we used fluorescence reporters, including GFP starting with CUG codons under a Kozak context and mCherry starting with AUG codons, to reflect the biases in initiation codon selection. The mRNAs encoding these two reporters were co-injected into *Bzw1*-knockdown zygote. Reporter assay showed that the relative levels of CUG initiation were higher in *Bzw1* RNAi 2-cell embryos (Fig. 6a, b). We also constructed a plasmid that Myc-protein starting with AUG codons under the Kozak context. The mRNAs encoding GFP starting with CUG codons and Myc starting with AUG were co-injected into the *Bzw1*-knockdown zygotes. The 130 embryos in each group were collected after in vitro culture for 48 h to conducted western blotting. Our results showed that the expression of CTG codon-initiated GFP was upregulated and that ATG codon-initiated Myc was downregulated in *Bzw1*-knockdown embryos. The ratio of the gray value of GFP to Myc was

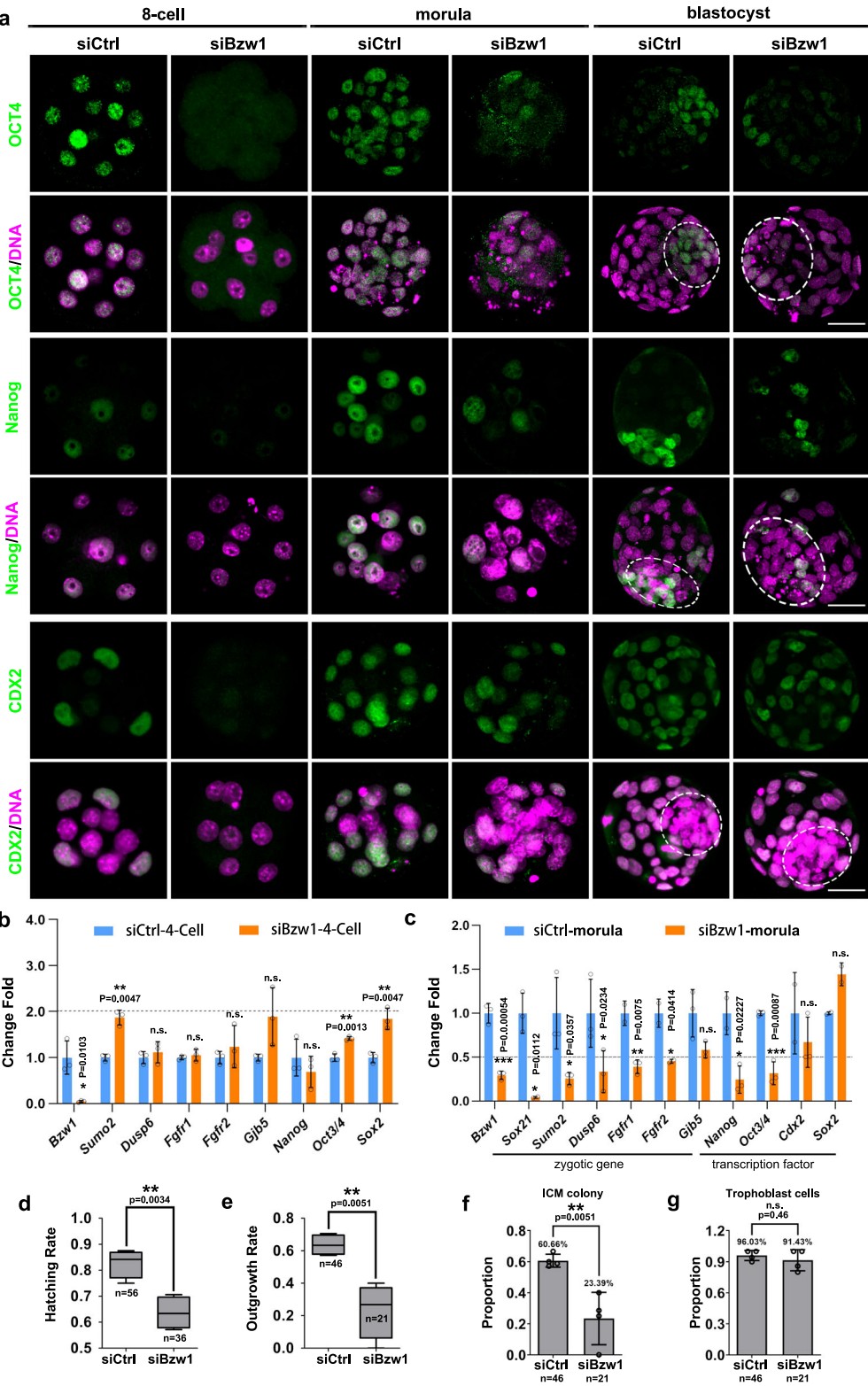

increased after *Bzw1* interference (Fig. 7c), which revealed that the lack of BZW1 would cause the choice of translation start codon tend to non-AUG codons. In addition, we explored the efforts of the tendency of translation start codon change. The efficiency of global protein synthesis after *Bzw1* knockdown was also tested. We incubated control and *Bzw1* RNAi embryos with L-homopropargylglycine (HPG), an analog of methionine that can be detected using the Click-iT HPG cell reaction kit. The results showed that *Bzw1*-knockdown embryos had weaker HPG signals than the control group at the 8-cell stage (Fig. 6d, e). OPP (O-propargyl-puromycin) assay also been conducted to detection of protein synthesis. Embryos from the 2–8-cell stages were incubated with OPP for 20 min, which was efficiently incorporated into newly translated proteins and fluorescently labeled with photostable Alexa Fluor™ dye. We found that *Bzw1*-knockdown embryos had weaker OPP signals than the control group at the 2–8-cell stages (Fig. 6f, g).

**Fig. 4 | BZW1 can regulate OCT4 and Nanog expression to ensure the differentiation capacity in preimplantation embryos. a** Immunofluorescence of CDX2, NANOG, and OCT4 (green) in si*Control* and si*Bzw1* embryos when the control embryos reached 8-cell, morula, and blastocyst stages. DNA was counterstained with DAPI (magenta). At least 20 embryos were observed in each experimental group and three independent experiments were conducted. Scale bar, 25 μm. **b** RT-PCR results showing relative mRNA levels of the indicated genes in embryos at the 4-cell stage after microinjection of siControl or si*Bzw1* in zygotes. *Gapdh* is served as the internal control. The relative mRNA level of *Bzw1* in the control group was defined as 1.0. Five embryos were collected for each sample. Data are presented as mean values. n.s. not significant, *$P < 0.05$, **$P < 0.01$, according to two-tailed Student's *t*-tests. Error bars, SD. **c** RT-PCR results showing relative mRNA levels of the indicated genes in embryos at the morula stage after microinjection of siControl or si*Bzw1* in zygotes. *Gapdh* is served as an internal control. The relative mRNA level of *Bzw1* in the control group was defined as 1.0. Data are presented as mean values.

Five embryos were collected for each sample. n.s. not significant, *$P < 0.05$, **$P < 0.01$, ***$P < 0.001$ by two-tailed Student's *t*-tests. Error bars, SD. **d** The proportion of blastocysts with siControl or si*Bzw1* microinjection hatched out from the zona pellucida. Data are presented as minima, maxima and mean values in the box plots. **$P < 0.01$, by two-tailed Student's *t*-test. Error bars, SD. **e** Proportions of blastocysts with siControl or si*Bzw1* microinjection that formed ICM colony with proliferating trophoblast cells. Data are presented as minima, maxima and mean values in the box plots. **$P < 0.01$, by two-tailed Student's *t*-test. Error bars, S.D. **f** The proportion of blastocysts that formed ICM colony in siCtrl and siBzw1 hatched out embryo. Data are presented as mean values. **$P < 0.01$, by two-tailed Student's *t*-test. Error bars, S.D. **g** The proportion of blastocysts whose trophoblast cells attached and grown in siCtrl and si*Bzw1* hatched embryo. Data are presented as mean values. n.s. not significant, by two-tailed Student's *t*-test. Error bars, S.D. All source data are provided as a Source Data file.

---

The tendency for mRNA translation initiation with non-AUG codons caused by decreases in BZW1 levels, may reduce translational efficiency during early embryonic development. Gene ontology (GO) analysis of all mRNAs with non-AUG start codons potentially affected by BZW1 in mouse early embryo development (1218 genes) revealed that key biological processes for embryo development were among the most enriched terms and biogenesis including translation, RNA splicing, ribonucleoprotein complex, and mitotic cell cycle process (Fig. 6e).

## Discussion

We have shown that the zygotic gene *Bzw1*, whose expression is specifically induced early in the zygotic stage, is essential for embryonic development. In particular, it plays an important role in embryo compaction (Fig. 8a). We also confirmed the mechanism by which BZW1 antagonizes the effects of eIF5 on non-AUG initiation to ensure normal translation initiation (Fig. 8a, b).

Several studies have reported that mRNA translation is controlled by cytoplasmic polyadenylation during the maternal-to-zygotic transition (MZT)[37–39]. However, there has been little research on the extent to which translational efficiency is affected by the stringency of start codon selection during zygotic genome activation. Here, we demonstrated that BZW1, an antagonist of eIF5 that improves non-AUG initiation, contributes to protein synthesis during early embryonic development. Some studies have shown that start codon recognition and selection are tightly controlled by multiple initiation factors[22,23,40]. However, the physiological function of this mechanism remains unclear. Overexpression of eIF5 increased non-AUG translation initiation in human HEK293T cells, which prompted us to increase non-AUG initiation in preimplantation embryos through eIF5 mRNA microinjection. The rate of compaction and blastocyst formation decreased after eIF5 overexpression (Fig. 5e, f), suggesting that disruption of the balance between AUG and non-AUG is detrimental to embryonic development. These phenotypes were similar to those observed with BZW1 depletion. However, we noticed that the degree of the effect differed between eIF5 overexpression and *Bzw1* RNAi. The latter was more serious for differentiation-related gene expression, embryonic compaction, and blastocyst development rate (Figs. 4a, 5e, f, h, i and S5c, d). This indicates that BZW1 has functions other than competing with eIF5 in early embryos.

Some studies have reported that BZW1 is highly expressed in cancer tissues and cell lines such as prostate cancer and lung adenocarcinoma[14,41,42]. BZW1 can promote cell proliferation in prostate cancer by regulating the transforming growth factor-beta1 (TGF-β1)/SMAD pathway[42]. Considering that some of the *Bzw1* RNAi embryos were arrested at the cleavage stage (Fig. 2c, d), we hypothesized that BZW1 may also play a role in regulating cell cycle and division during embryonic development. According to the ontology analysis (Fig. 7h)

of genes containing uTIS with non-AUG as the start codon and that were activated in early embryos, some proteins that regulated the mitotic cell cycle process can be BZW1 target gene candidates, including *Cdk1*[43], *Rad21*[44], and *Chek2*[45].

We also explored the relationship between *Bzw1* and its homologous gene *Bzw2*. Comparing si*Bzw1* and si*Bzw2* embryo development, it appeared that si*Bzw1* dominantly affected the compaction process, but BZW2 seemed to play more important role from the morula to blastocyst stages. Additionally, BZW1 and BZW2 can rescue the lack of each other by overexpression to some extent, suggesting a common effect in promoting embryo development. The different phenotypes may imply that their function has temporal specificity, which is also consistent with their expression patterns during preimplantation embryo development (Fig. 5a).

We selected candidate genes that contained uTIS with non-AUG as the start codon and that were activated in early embryos (Fig. S6a). Ontology analysis (Fig. 7h) of the chosen transcripts revealed that the protein expression of other translation regulation factors, including *Pabpc1*[39], *eIF1A*[46], and *Eef2*[47] may have been affected. This suggests that other steps of translation, such as poly(A)-tail lengthening, PIC assembly, or translation elongation, could also be impacted in *Bzw1*-knockdown embryos. Eventually, total protein levels were reduced. The selected gene *Rhoa*, which contains CTG and TTG start sites in uTIS, has been demonstrated to trigger polarization during early embryonic development[3]. The ligand fibroblast growth factor (*Fgf4*) receptor *Fgfr*1 contains CTG as the start codon. *Fgfr* genes are expressed in the ICM cells of blastocyst stage embryos[48]. *Fgfr1*−/− embryos are gastrulation defective, because *Fgfr1* plays a key role in primitive endoderm specification[48,49]. Although it is possible that the mRNA of many genes undergo non-AUG start codon translation initiation, the specific genes that increase non-AUG initiation of translation after *Bzw1* knockdown in early embryos need to be further explored.

The present study highlights the concept of stringent start codon recognition during zygotic gene protein translation activity, and supports early embryo development by ensuring precise protein synthesis (Fig. 8a). BZW1, as well as eIF5, plays key role in start codon selection during translation initiation. When these two proteins co-exist in embryos, they exhibit a competitive relationship in binding eIF2, which maintains the balance between AUG and non-AUC initiation (Fig. 8b). In agreement with our findings, a recent study showed that eIF1A knockdown by siRNA microinjection in mouse zygotes decreased the blastocyst formation rate[50]. Previous studies have also shown that eIF1A is a key factor in the accuracy of start codon selection both in vitro and in yeast model[23,46,51,52]. In clinical settings, some embryos have been shown to reach the 8-cell stage, but fail to form blastocysts. The expression level of BZW1/2 transcripts decreased significantly in 8-cell stage-arrested human embryos[53], suggesting that BZW1 has the

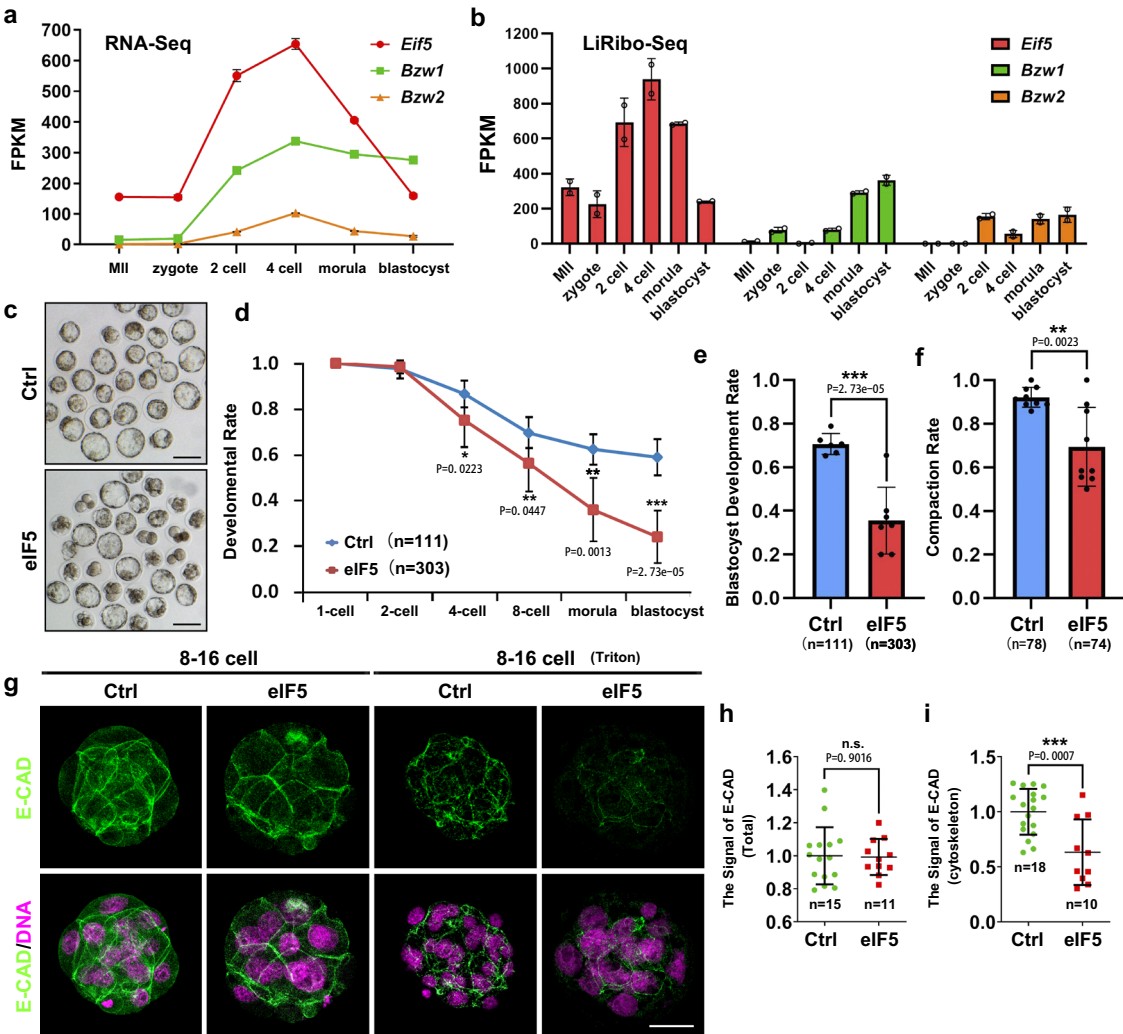

**Fig. 5 | Overexpression of eIF5 mediated non-AUG translation initiation rate increasing can mimic the phenotype that preimplantation embryonic development and compaction rate decreased causing by the lake of BZW1. a** The RNA-Seq from published data (GEO: GSE169632) shows the mRNA levels of *Bzw1, Bzw2,* and *eIF5* during early embryo development. *n* = 100–250 oocytes or embryos in every sample. Two replicates' data are presented as mean values, by two-tailed Student's *t*-test. Error bars, S.D. **b** Low-input ribosome profiling (LiRibo-seq) results from published data (GEO: GSE169632) showed the translational levels of *Bzw1, Bzw2,* and *eIF5* during early embryo development. *n* = 100-250 oocytes or embryos in every sample. Data are presented as mean values, by two-tailed Student's *t*-test. Error bars, S.D. **c** Representative embryo DIC images at E4.5. Zygotes were microinjected with mRNAs encoding mCherry (as a control) or mCherry-eIF5 (700 ng/μL) and cultured for 4 days. Scale bar, 100 μm. **d** Zygotes were microinjected with mRNAs encoding GFP or GFP-eIF5 (700 ng/μL) and then cultured for 4 days. The percentage of embryos that reached the 2-cell, 4-cell, 8-cell, morula, and blastocyst stages was calculated and is presented as the developmental rate. Data are presented as mean values. *$P$ < 0.05, **$P$ < 0.01, ***$P$ < 0.001, by two-tailed Student's *t*-

test. Error bars, SD. **e** Blastocyst developmental rates of embryos with overexpression of eIF5 as in **d**. Data are presented as mean values. ***$P$ < 0.001, by two-tailed Student's *t*-tests. Error bars, S.D. **f** The compaction rates of embryos with overexpression of eIF5 as in **d**. Data are presented as mean values. Error bars, SD. **$P$ < 0.01, by two-tailed Student's *t*-tests. **g** Confocal microscopic images of E-CAD (green) immunofluorescence in 8-cell to 16-cell stage embryos that were microinjected with eIF5 mRNAs at the zygote stage. The embryos have been treated with 1% Triton X-100 for 15 min before fixation to observe cytoskeletal protein (graph on the right). The DNA was counterstained with DAPI (magenta). Scale bar, 25 μm. **h** Signal intensities of E-CAD from **g** indicated the level of E-CAD in Control and eIF5-overexpressed embryos. Data are presented as mean values. n.s. not significant, by two-tailed Student's *t*-test. Error bars, S.D. **i** Signal intensities of E-CAD from **g** indicated the levels of E-CAD in control and eIF5-overexpressed embryos treated with 1% Triton X-100 before fixation. Data are presented as mean values. ***$P$ < 0.001, by two-tailed Student's *t*-test. Error bars, S.D. All source data are provided as a Source Data file.

potential to become a genetic marker or biomarker for early embryo quality and female infertility.

## Methods

Our research complies with all relevant ethical regulations. The Academic Committee and Ethics Committee of Reproductive and Genetic Hospital of CITIC-XIANGYA and Zhejiang University Institutional Animal Care and Research Committee (Approval # ZJU20170014), approved the study protocol.

### Mice strain, superovulation, and fertilization

Wild-type C57BL/6 mice were obtained from the Zhejiang Academy of Medical Science, China. Mice were maintained under specific pathogen free conditions in a controlled environment of 20–22 °C, with a 12/12 h light/dark cycle, 50–70% humidity, and food and water provided. For superovulation and fertilization, 4-week-old female mice were injected with 5 IU of pregnant mare serum gonadotropin (PMSG; Ningbo Sansheng Pharmaceutical Co., Ltd., China). After 44 h, the mice were injected with 5 IU human

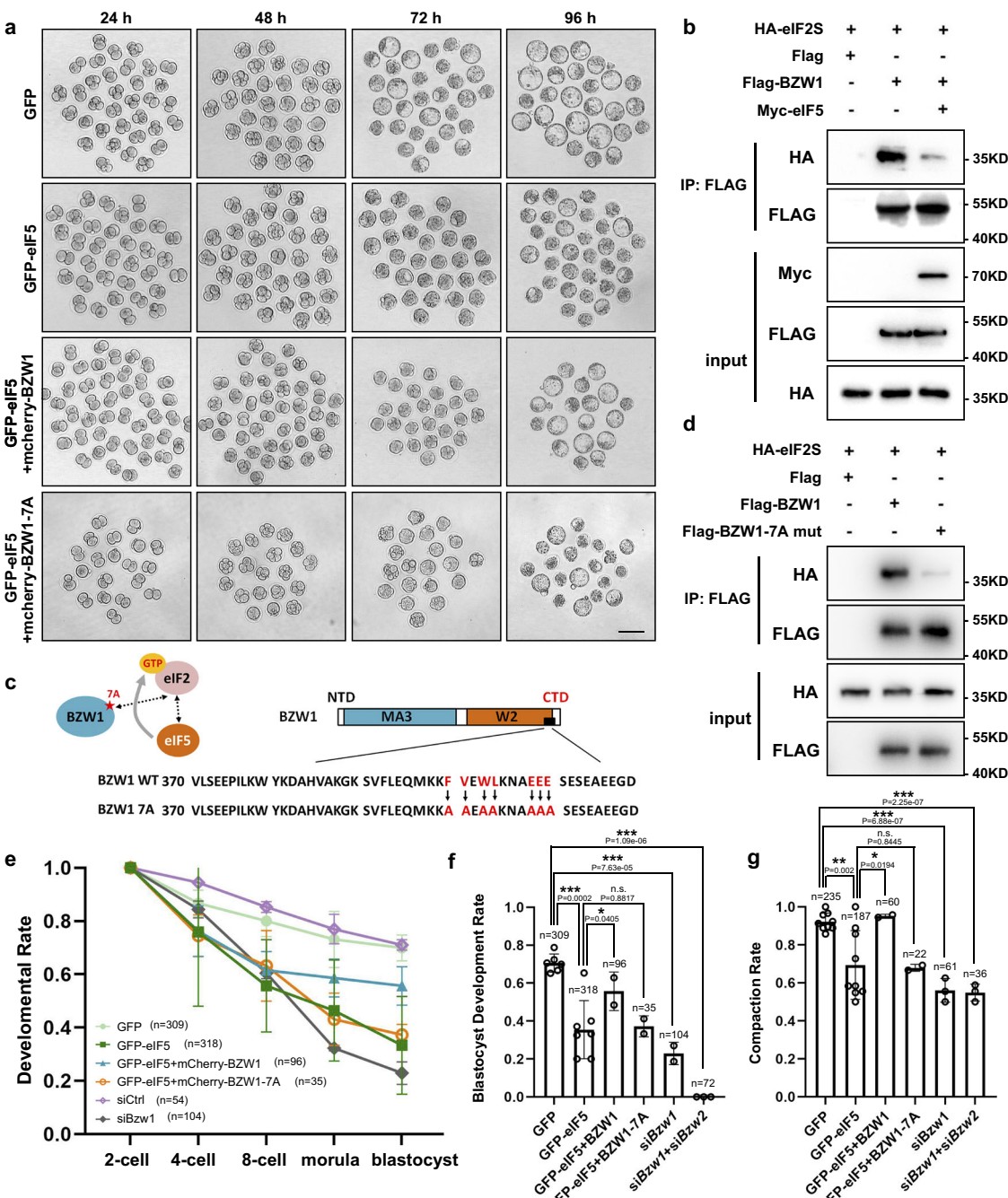

**Fig. 6 | BZW1 promotes preimplantation embryonic development and compaction by competing with eIF5. a** Representative embryo images of embryos at 24, 48, 72, and 96 h after microinjection in zygotes. These were respectively microinjected with mRNAs encoding GFP (as control), GFP-eIF5, GFP-eIF5 and mCherry-BZW1, or GFP-eIF5 and mCherry-BZW1-7A mutants. Scale bar, 100 μm. **b** Co-immunoprecipitation (Co-IP) experiments showing interactions of the eIF2 subunit with BZW1 with or without eIF5 overexpression. HeLa cells transiently transfected with plasmids encoding the indicated proteins were lysed and subjected to immunoprecipitation with an anti-FLAG affinity gel. Input cell lysates and precipitates were immunoblotted with antibodies against FLAG and HA. **c** Amino acid sequence in BZW1-7A mutant CTD. The experiment was repeated two times with similar results. **d** Co-IP experiment demonstrating interactions of the eIF2 subunit with BZW1 or BZW1-7A mutants. The experiment was repeated three times with similar results. **e** Developmental rates of embryos that reached every stage in six groups with microinjection of mRNAs as in **a** and indicated siRNAs. The number of embryos was counted at 24, 48, 72, and 96 h, respectively. Data are presented as mean values. Error bars, SD. **f** Embryos microinjected with mRNAs as in **a** or indicated siRNAs cultured for 4 days in vitro. The percentage of embryos that reached the blastocyst stage was calculated and it is presented as the developmental rate. Data are presented as mean values. n.s. not significant, *P < 0.05, ***P < 0.001, by two-tailed Student's t-test. Error bars, S.D. **g** Embryos microinjected with mRNAs as in **a** or siBzw1 were cultured for 4 days in vitro. The percentage of embryos that completed compaction was calculated and it is presented as the compaction rate. Data are presented as mean values. n.s. not significant, **P < 0.01, ***P < 0.001, by two-tailed Student's t-test. Error bars, S.D. All source data are provided as a Source Data file.

chorionic gonadotropin (hCG; Ningbo Sansheng Pharmaceutical Co., Ltd., P. R. China) and mated with adult males. Successful mating was confirmed based on the presence of a vaginal plug. The experimental protocols that involved mice were approved by the

Zhejiang University Institutional Animal Care and Research Committee (Approval # ZJU20170014), and mouse care and use was performed in accordance with the relevant guidelines and regulations.

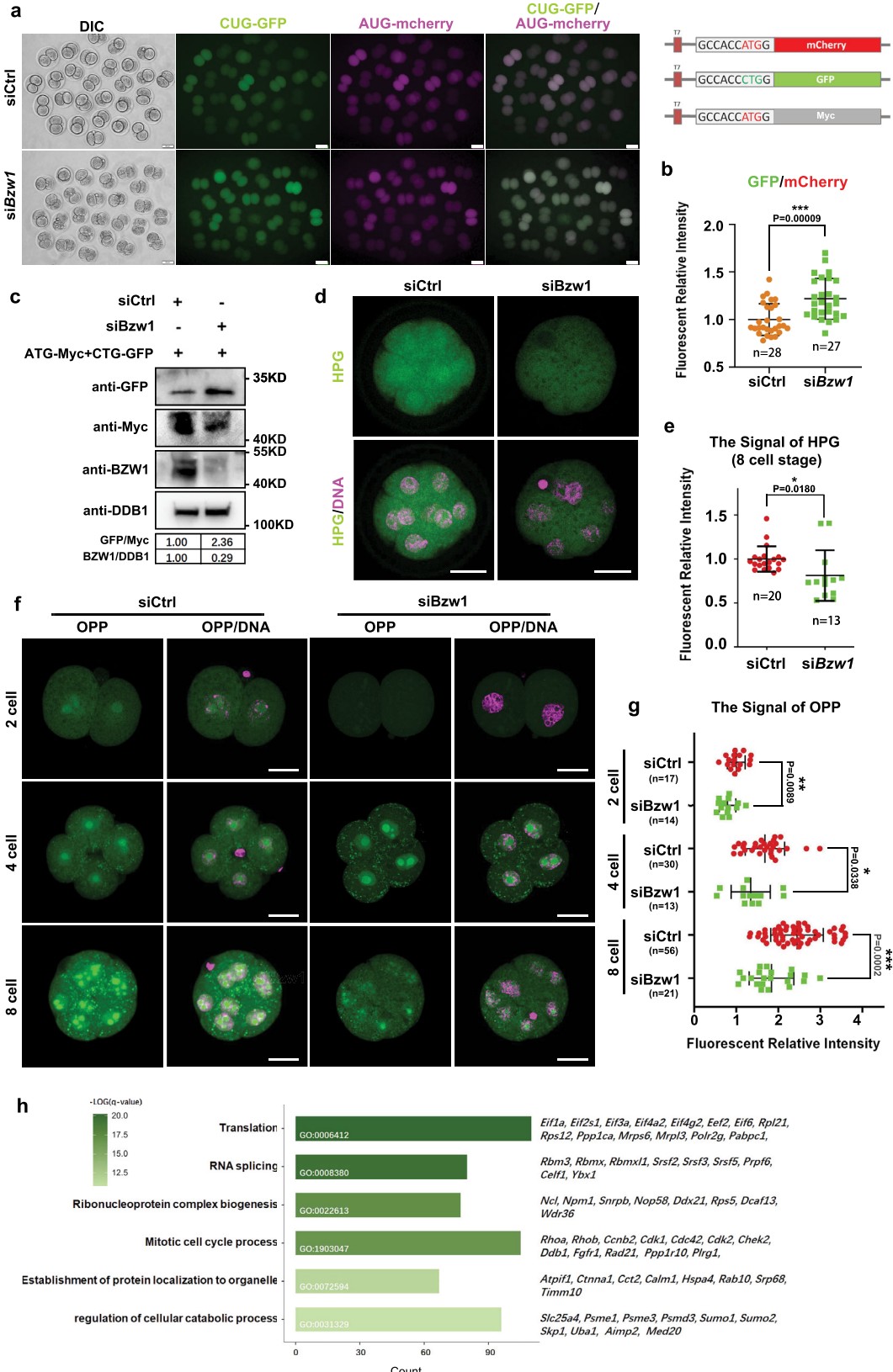

## Zygote collection, in vitro culture and microinjection

Mouse zygotes were obtained from oviducts by inducing super-ovulation in 4-week-old females that were mated with males of the same strain. Zygotes were released in 37 °C pre-warmed M2 medium (Merck, M7167, Germany) and cultured in mini-drops of KSOM (Merck, MR-106; Germany) covered with mineral oil (Sigma–Aldrich, M5310; USA) at 37 °C in a 5% $CO_2$ atmosphere. Embryos that reached the 2-cell, 4-cell, 8-cell, morula, and blastocyst stages were obtained at 36, 48, 60, 72, and 84 h post-hCG injection. During in vitro embryo culture, we transported the embryos into the fresh culture medium every 48 h.

For microinjection, zygotes were collected in the M2 medium covered with mineral oil. mRNAs were transcribed in vitro using the

**Fig. 7 | Decreased BZW1 level leads to increased non-AUG initiation rate in preimplantation embryos. a** Fluorescence microscopy exhibiting the expression of GFP with a CUG start codon and mCherry with an AUG start codon in 2-cell embryos microinjected with control and *Bzw1* siRNA at the zygote stage. Scale bar, 50 μm. **b** Relative fluorescence intensity indicates the percentage of CUG initiation compared with AUG initiation in **a**. Data are presented as mean values. n indicates the number of embryos were analyzed. ***P < 0.001, by two-tailed Student's *t*-tests. Error bars indicate SD. **c** Western blotting showing the level of GFP with a CUG start codon and Myc with an AUG start codon in embryos after microinjection with siCtrl and si*Bzw1* 36 h. DDB1 was blotted as a loading control. Total protein samples from 130 embryos were loaded into each lane. Sizes (kDa) of the protein markers are indicated on the right. The gray values of GFP were normalized to Myc. The gray value of BZW1 was normalized to DDB1. The experiment was repeated two times with similar results. **d** HPG fluorescent staining results showing protein synthesis levels of embryo in 8-cell stage embryo. Control and *Bzw1* siRNA microinjected embryos were incubated in no methionine medium containing 50 μM HPG for 30 min prior to staining. At least 20 embryos were stained for each stage. Scale bars,

25 μm. **e** Quantification of HPG signal intensity in **d**. HPG signal was normalized against the average signal intensity in the siCtrl group. Data are presented as mean values. *n* indicates the number of embryos identified and analyzed for each group. *P < 0.05, by two-tailed Student's *t*-test. Error bars, S.D. **f** OPP fluorescent staining results showing protein synthesis levels of embryo from 2-cell to 8-cell stage. The siCtrl and si*Bzw1* microinjected embryos were incubated in M16 medium containing 50 μM OPP for 20 min prior to staining. At least 10 embryos were stained at each stage. Scale bars, 25 μm. **g** Quantification of OPP signal intensity in **f**. OPP signal was normalized against the average signal intensity in the 2-cell siCtrl group. Data are presented as mean values. n indicates the number of embryos identified and analyzed for each group. *P < 0.05, **P < 0.01, ***P < 0.001, by two-tailed Student's *t*-test. Error bars indicate SD. **h** Gene ontology (GO) analysis of potentially affected mRNA. The bar chart shows the −log10-transformed significance value and representative genes of the GO terms in the described biological processes. GO analyses of the candidate genes were performed using Metascape (http://metascape.org/). All source data are provided as a Source Data file.

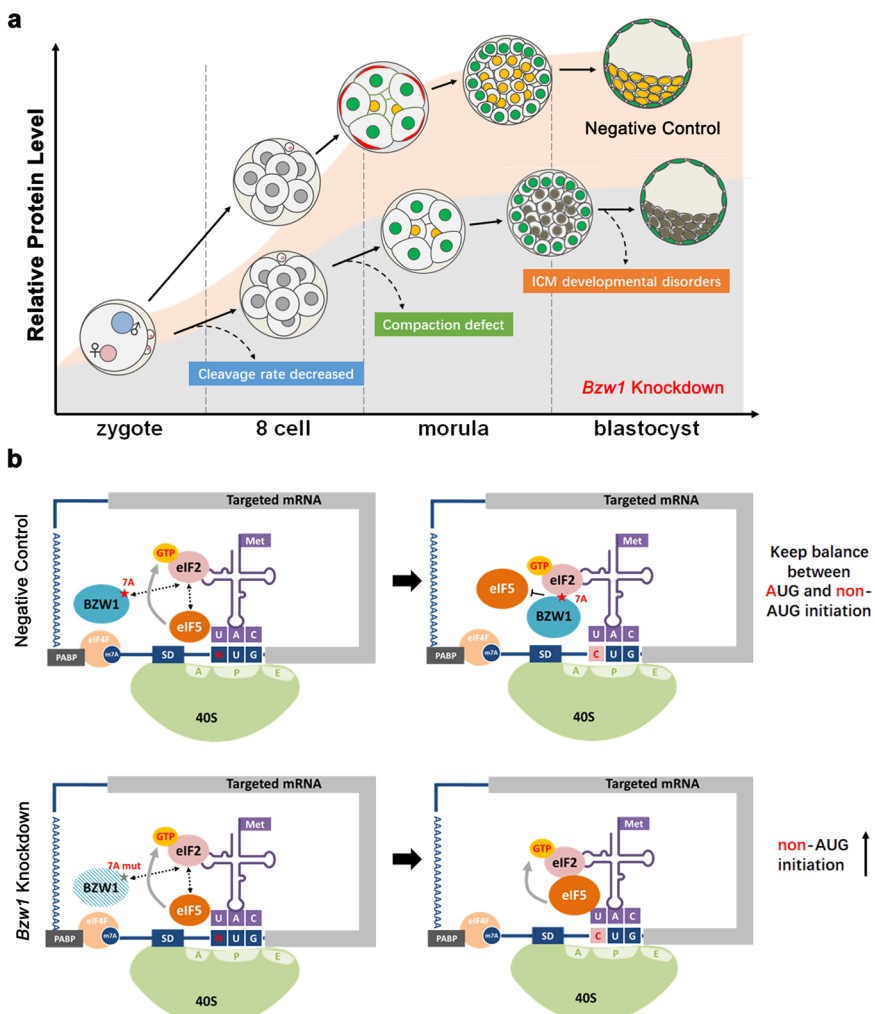

**Fig. 8 | BZW1 can regulate preimplantation embryonic development and total protein levels by restricting non-AUG translation initiation. a** BZW1 is required for ensure total protein synthesis during early embryonic development. The lack of BZW1 decreased the rate of early embryo cleavage. The *Bzw1* RNAi embryos that reached the 8-cell stage failed to develop into the morula stage. A small proportion of *Bzw1*-knockdown embryos developed to the blastocyst stage, but with abnormal

ICM differentiation. **b** BZW1 and eIF5, play key roles in start codon selection during cytoplasmic translation initiation. When these two proteins co-exist in embryos, they have a competitive relationship in binding eIF2, which keep the balance between AUG and non-AUC initiation. When BZW1 was insufficient, excessive eIF5 interacted with PIC, the rate of non-AUG translation initiation increased and protein accumulation efficiency decreased.

mMESSAGE mMACHINE™ SP6 Transcription Kit (Invitrogen, AM1340, USA) or mMESSAGE mMACHINE™ T7 Transcription Kit (Invitrogen, AM1344, USA). *Bzw1* siRNA was synthesized by RiboBio biotechnology company (Beijing, China). Microinjection was performed using a

micromanipulator and microinjector (Eppendorf) and an inverted microscope (Eclipse TE200; Nikon). Approximately 10 pL synthetic mRNA (800 μg/mL) or siRNA (20 μM) diluted in distilled water was microinjected into the cytoplasm of oocytes.

## Outgrowth assay

Blastocysts were harvested and cultured individually in a single well of DMEM (Lonza, Allendale, NJ, USA) containing 10% fetal bovine serum. Blastocysts were allowed to attach and outgrow for three days at 37 °C in a 5% $CO_2$ atmosphere. Outgrowths were evaluated according to morphology as previously researches[54].

## Western blotting

In vivo zygotes were collected from the oviducts, and cultured for 3.5 days in vitro. Embryos that reached the 2-cell, 4-cell, 8-cell, morula, and blastocyst stages were collected at 36, 48, 60, 72, and 84 h post-hCG injection. These embryos were lysed directly in β-mercaptoethanol containing loading buffer and heated at 95 °C for 10 min. Sodium dodecyl sulfate polyacrylamide gel electrophoresis (SDS-PAGE) and immunoblotting was performed following standard procedures using a Mini-PROTEAN Tetra Cell System (Bio-Rad, Hercules, CA, USA). The antibodies used are listed in Supplementary Table 1.

## Single-embryo RT-PCR

This method was modified from the previously reported RNA Smart-seq2 protocol[55]. Every five embryos were lysed in 2 µL lysis buffer (0.2% Triton X-100 and 2 IU/µL RNase inhibitor), serving as a sample. The sample was reverse transcribed using the SuperScript III reverse transcriptase (Thermo Fisher, #18080085, MA, USA) and amplified by PCR for 10 cycles. PCR products were diluted and used as templates for RT-PCR.

Quantitative RT-PCR was performed using the Power SYBR Green PCR Master Mix (Applied Biosystems, Life Technologies, USA) with an ABI 7500 Real-Time PCR system (Applied Biosystems, Life Technologies, USA). Relative mRNA levels were calculated by normalizing to the endogenous GAPDH mRNA levels. The primer sequences used are listed in Supplementary Table 2 and Table 3.

## Single-embryo RNA-sequencing (RNA-seq)

Embryo mRNA libraries were sequenced on a BGISEQ500 platform (BGI-Shenzhen, China) with 100 bp paired-end reads. All reads that passed through the filter were trimmed to remove low-quality bases and adaptor sequences by fastp (v0.20.0). Reads were aligned to the mouse reference genome of GRCh38 using STAR (v2.7.8a) and transcripts per million (TPM) were calculated and normalized using StringTie (v2.0.4). Differentially expressed genes were analyzed using the R package DESeq2 (filtered with q-value ≤ 0.01 and Foldchange ≥ 2).

## Cell culture and plasmid transfection

HeLa cells (ATCC, USA) were authenticated and tested for mycoplasma contamination. HeLa cells were grown in DMEM (Invitrogen) supplemented with 10% fetal bovine serum (HyClone) and 1% penicillin–streptomycin solution (Gibco) at 37 °C in a humidified 5% $CO_2$ incubator. HeLa cells were transfected with the plasmids using Lipofectamine 3000 (Invitrogen). HeLa cells were cultured for 24 h for immunofluorescence staining and for 48 h for co-immunoprecipitation.

## Immunofluorescent staining

Embryos or HeLa cells were fixed with 4% paraformaldehyde in phosphate-buffered saline (PBS) at room temperature for 30 min. The cells were then permeabilized with 0.3% Triton X-100 in PBS for 40 min. Some embryos were treated with 1% Triton X-100 (Sigma T8787) for 15 min before fixation to observe the proteins localized on the cytoskeleton. Antibody staining was performed using standard protocols described previously[56]. The antibodies used in these experiments are listed in Supplementary Table 1. The embryos were performed were imaged using a Zeiss LSM710 confocal microscope. The image represents the maximum intensity projection of the z-stack.

Semi-quantitative analysis of the fluorescence signals was performed using ImageJ or ZEN 2 imaging software.

## Detection of protein synthesis

Embryos were incubated in no methionine DMEM (#21013024, Thermo Fisher, MA, USA) supplemented with 100 mM HPG for 30 min. followed by fixation for 30 min at 37 °C in 3.7% formaldehyde. HPG was detected using Click-iT™ HPG Alexa Fluor™ 488 (#C10428, Thermo Fisher, MA, USA). The mean signal was measured, crossed the middle of each embryo, and quantified using ImageJ software.

Embryos were incubated in M16 (Sigma–Aldrich, M7292, USA) supplemented with 100 mM OPP (O-propargyl-puromycin) for 20 min. The embryos were fixed for 30 min at 37 °C in 3.7% formaldehyde. OPP was detected using Click-iT™ Plus OPP Alexa Fluor™ 647 Protein Synthesis Assay Kit (#C10458, Thermo Fisher, MA, USA).

## Immunoprecipitation

To detect the interaction between BZW1 and other eIFs, the plasmids of interest were co-transfected into HeLa cells using Lipofectamine 2000 (Invitrogen). Forty-eight hours after transfection, cells were lysed in lysis buffer (50 mM Tris-HCl, pH 7.5, 150 mM NaCl, 10% glycerol, and 0.5% NP-40; protease inhibitors were added prior to use). After centrifugation at $12,000 \times g$ for 30 min, the supernatant was subjected to immunoprecipitation with the ANTI-FLAG M2 Affinity Gel (Sigma, A2220) for 4 h at 4 °C. The immunoprecipitated beads were washed three times with the lysis buffer. SDS sample buffer was added to the beads and supernatant (input) and the eluates were used for western blot analysis.

## Statistical analysis

Results are presented as the mean ± standard deviation (SD). Each experiment was repeated at least three times. The results for two experimental groups were compared using two-tailed unpaired Student's $t$-tests. Statistically significant values of $P < 0.05$, $P < 0.01$, and $P < 0.001$ are indicated by asterisks (*), (**), and (***), respectively. GO analyses were performed using the open gene enrichment analysis tool, Metascape (http://metascape.org/).

## Reporting summary

Further information on research design is available in the Nature Research Reporting Summary linked to this article.

## Data availability

The single-cell RNA-seq raw data generated in the manuscript are available in the Sequence Read Archive (SRA) database under the accession code PRJNA781008. Source data are provided with this paper.

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

## Acknowledgements

This research was supported by the National Natural Science Foundation of China (31930031, 31890781 to H.Y.F.), Excellent Postdoctoral Talents Innovation Project in Hunan Province (2020RC2050 to J.Z.), China Postdoctoral Science Foundation (2021M690983 to J.Z.), and the Natural Science Foundation of Zhejiang Province, China [LD22C060001] to H.Y.F.

## Author contributions

H.Y.F and L.G. supervised the project. H.Y.F., L.G., and J.Z. designed the experiments. J.Z., S.P., and J.G. performed the embryo microinjections, manipulations, and staining experiments. S.P. and N.Z. performed the molecular experiments. J.Z., W.Z., and L.L. performed the statistical analyses. H.Y.F. and J.Z. wrote the manuscript.

## Competing interests

The authors declare no competing interests.
