## [Peer Review File · Nature Communications]

Translation Regulatory Factor BZW1 Regulates
Preimplantation Embryo Development and Compaction by
Restricting Global non-AUG InitiationREVIEWER COMMENTS

Reviewer #1 (Remarks to the Author):

In this manuscript, the authors investigated the role of BZW1 in protein translation control (translation start codon selection) in early embryo development. They performed knockdown experiments at zygote stage then they observed that the compaction of early embryos were greatly affected and also the blastocyst development is significantly decreased. Subsequently, they analyzed the transcription and translation of the embryos, and they concluded that the knockdown of BZW1 mainly affected the translation start codon AUG selection. Overall, the experiments were well designed and conducted, and the results seem interesting and important for the field, because there are very few studies focusing on the protein translation control in early embryos.

There are some major concerns need to be clarify or further prove by doing additional experiments or analysis.

1. In the first part of the study, the authors performed knockdown experiments and showed that loss of function of BZW1 can cause decreased blastocyst development, then they further showed that the expression of Oct4 and Nanog was down-regulated. But, the authors mentioned that the gene expression is not altered in embryos when BZW1 was knocked down. How to interpret this observation?
2. Related to question 1, the authors also found that E-CAD is greatly reduced in BZW1 knockdown embryos, is it because the selection of translation start codon altered in E-CAD mRNA?
3. In this manuscript, the authors mainly used knockdown method to investigate the role of BZW1 in early embryo development. What is the phenotype when BZW1 is knockout? Do the KO embryos die at a specific time point?
4. In BZW1-knockdown embryos, the polarity of the blastomeres was affected? This reviewer guesses that the blastomeres will lose polarity in the embryos with no compaction. Then the following question will be how the Cdx2 expression is unregulated?

Reviewer #2 (Remarks to the Author):

The paper by Zhang et al. reports the essential roles of BZW1, a translation regulatory factor, in the synthesis of proteins necessary for periimplantation development of early mouse embryos. BZW1 exerts its effect on the protein synthesis by restricting non-AUG translation initiation via inhibition of eIF5, as revealed by the authors' comprehensive biochemical as well as inhibition/rescue experiments using mouse embryos. The experiments were well-designed, the results were clearly presented, and final conclusions were logically drawn from the experimental results obtained. Overall, the paper is of a highly scientific quality, and will attract broad interests of researchers in the developmental biology as well as biochemistry. However, there are some points to be improved so that the paper can be a stronger and more citable paper in the related fields.

Major points

1. The authors showed that Bzw1 was more highly expressed in mouse embryos, especially during 2-cells to 4-cells, than Bzw2. This is the main reason why the authors focused on BZW1, not BZW2. However, unlike siRNA treatment for Bzw1 alone, the combination of two siRNAs for Bzw1 and Bzw2, completely arrested development of embryos before the blastocyst stage (Fig. S3B, D). This result indicates that Bzw2 also plays a significant role in preimplantation embryonic development. Therefore, I strongly recommend the authors to add at least the effect of siBzw2 on embryonic development to Fig. 1. Then, with the comparison of siBzw1 and siBzw2, the authors will be able to rationally explain why they focused on Bzw1. A discussion of the expected division of roles between Bzw1 and Bzw2 will also be necessary.
2. Knockdown of Bzw1 caused decreased developmental rates as early as during 4-cells and 8-cells. Therefore, there should have been some adverse effects of Bzw1 knockdown on the physiology of

embryos at these stages. Although the Bzw1 knockdown effect on compaction is interesting and developmentally important, there is no reason to ignore this earlier embryonic arrest. The authors should mention these earlier effects of Bzw1 knockdown at least in the Discussion.

Minor points

1. Fig. 1B: The BXW1 band at 4-cell seems to be at a lower position and associated with a small band. Is this any artifact? Or is this true?
2. Fig. 3C: This figure needs further explanations. Probably, the mCherry-positive areas of the "21/31" embryo may be important... The authors should explain the points to be focused on in the text or figure legend.
3. Page 6, line 26: What are the "differentiation markers"? Is this terminology correct?
4. Page 7, line 13: Hatching of blastocysts largely depends on the activity of trophoctoderm. However, the ICM was mostly affected in this case. Please explain this discrepancy.
5. Fig. S2F: What is "ERM" in the Y axis? This figure shows the p-Ezrin intensity.
6. Fig. S2G: I recommend this figure to be associated with Fig. 5G.
7. Page 9, line 12 "In addition, ...": Please check the composition of this sentence.

Reviewer #3 (Remarks to the Author):

In their MS entitled: Translation Regulatory Factor BZW1 Regulates Preimplantation Embryo Development and Compaction by Restricting Global non-AUG Initiation, Zhang and colleagues investigate the role of a translational regulator Bzw1 on preimplantation mouse development. Knockdown of this factor in the zygote leads to developmental defects from 8-cell stage, with reduced rates of blastocyst formation, compaction and aberrant morphologies. The authors claim that Bzw1 is necessary for morula compaction and pluripotency factor expression, and also that it regulates global protein synthesis as early as the 2-cell stage through restricting non-AUG translation initiation. While the developmental phenotype with reduced blastocyst formation seems to be consistent and merits further investigation, the direct experimental evidence in this MS does not support some of the claims laid out by the authors. Importantly, without additional global molecular profiling of embryonic transcription (RNA-seq) or translation, in my view it is impossible to conclude on the origin of the developmental phenotype from the data present in the MS.

I outline some of my questions below.

Fig2: What is the effect of Bzw1 KD on Bzw2 RNA and/or protein levels? Is there functional compensation? Fig 2C: There seems to be a decrease in developmental rate already between 4 and 8-cell stage of development (significantly lower number of 8-cell stage embryos), which occurs much prior to compaction and cell lineage allocation. Have the authors considered that the global phenotype later in development arises due to an earlier developmental defect, and not compaction? Secondly, culturing unhealthy embryos together with potentially healthy ones will certainly affect the developmental rates of the otherwise perfectly fine embryos.

Figure 3: What is the rationale behind injecting 1 of the 2 embryonic blastomeres? The authors write this is to reduce Bzw1 expression in half of the embryos (I suspect this meant embryo), but they then don't look at biases in cell fate allocation. I am unclear as to the conclusions drawn from these perturbations. Looking at E-Cad levels in siCtrl and siBzw1 embryos in Figure 3C, I would have a hard time seeing much difference if the figure wasn't annotated.

Figure 4: Do immunofluorescence panels show individual Z-slices or maximal projection images? In any case, there are lower levels of both proteins and RNA, and not only Oct3/4, Nanog and Cdx2 are mis regulated but apparently numerous other genes at an RNA and protein level. Later the authors

indicate that P-RNAPII levels are fine (which they conclude indicates that transcription is unaffected). What is causing then the downregulation of these indicated genes at the RNA level? Furthermore, could the authors please clarify Figure 4E?

Figure 5: Figure 5C shows embryos at different developmental stages (control is a blastocyst and the eIF5 OEx is an 8-cell/morula). Since many eIF5 OEx embryos make it to the blastocyst stage (as per Figure 5A), could the authors show E-Cad levels for developmentally matched embryos? Fig 5E I am somewhat confused as to why third row panel (rescue experiment) shows markedly fewer embryos in the 72h and 96h timepoints compared to 24h and 48h. Can the authors comment?

What are the relative endogenous levels of eIF5 during preimplantation development and how do they compare to Bwz1 levels? What is the direct evidence that Bwz1 competes with eIF5? The authors could try an in vitro competition assay with a cognate RNA and Bwz1 and eIF5.

Fig6: General comment: there is a lot of variability between fluorescence levels in both embryo groups, since this depends on how much mRNA was injected into each embryo, and many other effects (fluorescence bleaching, pH, position of the embryo, etc). Concluding of the efficiency of codon usage based on single plane fluorescent image quantitation of ~30 embryos is not exactly precise molecular data. Furthermore, in figure 6C and D, the authors claim there is a global reduction in translation / protein synthesis upon Bwz1 KD which is certainly not obvious from looking at GFP or mCherry signals in panel 6A. In 6C, it also appears that 2 of the 4 blastomeres of the 4-cell stage embryo there is higher HPG signal. Is this biological or technical? Have the authors attempted to do an OPP incorporation assay? In 6C, authors write: at least 20 embryos were analyzed per stage. In 6D authors show quantitation of 20 control and only 13 siBwz1 embryos. Why is that?

Re: # NCOMMS-22-52100A

Point-to-Point Responses to Reviewers' Comments

Reviewer #1 (Remarks to the Author):

In this manuscript, the authors investigated the role of BZW1 in protein translation control (translation start codon selection) in early embryo development. They performed knockdown experiments at zygote stage then they observed that the compaction of early embryos were greatly affected and also the blastocyst development is significantly decreased. Subsequently, they analyzed the transcription and translation of the embryos, and they concluded that the knockdown of BZW1 mainly affected the translation start codon AUG selection. Overall, the experiments were well designed and conducted, and the results seem interesting and important for the field, because there are very few studies focusing on the protein translation control in early embryos. There are some major concerns need to be clarified or further prove by doing additional experiments or analysis.

Response: We thank the reviewer for carefully reviewing our manuscript and for his/her supportive comments that the experiments were well designed and conducted, and the results are important for the field. In particular, we appreciate the reviewer's recognition that there is limited knowledge about protein translation control during early embryo development. We have carefully addressed the reviewer's comments and suggestions, as detailed below.

1. In the first part of the study, the authors performed knockdown experiments and showed that loss of function of BZW1 can cause decreased blastocyst development, then they further showed that the expression of Oct4 and Nanog was down-regulated. But, the authors mentioned that the gene expression is not altered in embryos when BZW1 was knocked down. How to interpret this observation?

Response: We appreciate the reviewer's comments. In the original Figure S4D (revised Figure S6C), we showed the transcriptional change by single-cell RNA-seq in *siBzw1* at 4-cell stage. The results indicated that 98.5% of the detected genes were almost unchanged (< 2 change folds) after *Bzw1* knockdown. This suggests that the decrease in BZW1 had no significant effect on transcription in the early stages after BZW1 knocked down. However, with embryo development, the degree of translation defects became increasingly serious. The embryos showed abnormal morphology and the expression of some ZGA genes was down-regulated at the morula stage. To clarify this, we added the ZGA gene expression level at the 4-cell and morula stages tested by RT-PCR in the revised Figure 4B.

2. Related to question 1, the authors also found that E-CAD is greatly reduced in BZW1 knockdown embryos, is it because the selection of translation start codon altered in E-CAD mRNA?

Response: We examined the localization of E-cadherin in the cytoskeleton to assess filopodia

formation. Treatment of cells with 1% Triton X-100 dissolved the membranes and soluble components, leaving intact cytoskeletal residues. The embryos were treated with 1% Triton X-100 for 15 min before fixation to observe the cytoskeletal protein. We apologize for not providing the total level of E-CAD in the original manuscript. We added E-CAD immunofluorescence before Triton X-100 treatment in Figure 3C-D to show the total E-CAD in the embryo. These results indicated that while the total expression level of E-cadherin was not affected by *Bzw1* depletion, the cytoskeletal attachment ability of E-cadherin was reduced in *Bzw1* knockdown embryos.

3. In this manuscript, the authors mainly used knockdown method to investigate the role of BZW1 in early embryo development. What is the phenotype when BZW1 is knockout? Do the KO embryos die at a specific time point?

Response: According to the data of the International Mouse Phenotyping Consortium, BZW1 KO mice survive into adulthood, but show abnormal auditory brainstem response, lens opacity and male infertility (<https://www.mousephenotype.org/data/genes/MGI:1914132>). It is common that the RNAi depletion and knockout of an early zygotic gene have different phenotypes. There are two possibilities: 1) The genetic compensation response has recently been proposed as a possible explanation for the phenotypic discrepancies between gene-knockout and gene-knockdown (Ma Z et al., Nature. 2019, 568: 259-263. doi: 10.1038). These phenotypic differences have been attributed to the upregulation of other genes in the same families in KO embryos, such as *Bzw2*. In RNAi experiments. However, *Bzw1* was promptly depleted so that the embryos did not have time to activate the mechanism that compensates for the loss of *Bzw1* (revised Fig. 2D); 2) It is known that mammalian embryos developed *in vivo* have a stronger ability to adjust cell fates than those developed *in vitro*. Therefore, the effect of *Bzw1* loss was more clearly demonstrated by *in vitro* developed embryos than *in vivo*.

4. In BZW1-knockdown embryos, the polarity of the blastomeres was affected? This reviewer guesses that the blastomeres will lose polarity in the embryos with no compaction. Then the following question will be how the Cdx2 expression is unregulated?

Response: BZW1-knockdown embryos that lose polarity with no compaction cannot develop into the morula and blastocyst stage. The embryos that showed CDX2 levels in Figure 4 have reached the morula and blastocyst stages. We further emphasized the phenotype of BZW1-knockdown embryos in the main text. To clarify the effect of BZW1 on the polarity of blastomeres, *Bzw1* siRNAs were microinjected into one blastomere of 2-cell stage embryos. The polarity of the blastomeres was observed by pEzrin staining. The pEzrin^{T567} was detected at the early 8-cell embryo stage, which mediates the formation of F-actin at the apical surface, and promotes embryo compaction and polarization. During apical protein polarization, pEzrin^{T567} concentrated at the center of the cell-contact-free surface to form an apical patch (revised Figure 3F, G-a). But pEzrin signal in the blastomeres with *Bzw1* knockdown was uniform

distributed on the cell-contact-free surface (revised Fig 3F, G-b). The pEzrin signal intensity in half *Bzw1* knockdown embryos was lower than that in control embryos (revised Figure 3I).

Reviewer #2 (Remarks to the Author):

The paper by Zhang et al. reports the essential roles of BZW1, a translation regulatory factor, in the synthesis of proteins necessary for periimplantation development of early mouse embryos. BZW1 exerts its effect on the protein synthesis by restricting non-AUG translation initiation via inhibition of eIF5, as revealed by the authors' comprehensive biochemical as well as inhibition/rescue experiments using mouse embryos. The experiments were well-designed, the results were clearly presented, and final conclusions were logically drawn from the experimental results obtained. Overall, the paper is of a highly scientific quality, and will attract broad interests of researchers in the developmental biology as well as biochemistry. However, there are some points to be improved so that the paper can be a stronger and more citable paper in the related fields.

Response: We thank the reviewer for carefully reviewing our manuscript and his/her supportive comment that the key conclusions are well supported by the data in the manuscript. We have carefully addressed the reviewer's concerns as detailed below.

Major points

1. The authors showed that *Bzw1* was more highly expressed in mouse embryos, especially during 2-cells to 4-cells, than *Bzw2*. This is the main reason why the authors focused on BZW1, not BZW2. However, unlike siRNA treatment for *Bzw1* alone, the combination of two siRNAs for *Bzw1* and *Bzw2*, completely arrested development of embryos before the blastocyst stage (Fig. S3B, D). This result indicates that *Bzw2* also plays a significant role in preimplantation embryonic development. Therefore, I strongly recommend the authors to add at least the effect of si*Bzw2* on embryonic development to Fig. 1. Then, with the comparison of si*Bzw1* and si*Bzw2*, the authors will be able to rationally explain why they focused on *Bzw1*. A discussion of the expected division of roles between *Bzw1* and *Bzw2* will also be necessary.

Response: We thank the reviewer for the helpful suggestion. We have added the effect of si*Bzw2* on embryonic development to the revised Figure 2. We also showed that BZW1 overexpression partially rescued the developmental defect of si*Bzw2* embryos; BZW2 overexpression, in turn, could also rescue the developmental defect of si*Bzw1* embryos, suggesting that there are functional compensations between BZW1 and BZW2 to some extent. We have provided this data in revised Figure 2 and S2. By comparing the si*Bzw1* and si*Bzw2* embryos development, we found out that si*Bzw1* affected the compaction process, but BZW2 seemed to play a more important role during morula to blastocyst development. We believed that this is because they have different expression patterns during embryonic development (Figure 5B). We have added this point to the revised Discussion.

2. Knockdown of Bzw1 caused decreased developmental rates as early as during 4-cells and 8-cells. Therefore, there should have been some adverse effects of Bzw1 knockdown on the physiology of embryos at these stages. Although the Bzw1 knockdown effect on compaction is interesting and developmentally important, there is no reason to ignore this earlier embryonic arrest. The authors should mention these earlier effects of Bzw1 knockdown at least in the Discussion.

Response: Indeed, some of the *Bzw1* RNAi embryos were arrested in the cleavage stage (revised Fig. 2C, D). We hypothesized that BZW1 may also play a role in regulating the cell cycle and division during embryonic development. Some studies have reported that BZW1 is highly expressed in cancer tissues and cell lines such as prostate cancer and lung adenocarcinoma. BZW1 promotes cell proliferation in prostate cancer by regulating the transforming growth factor-beta1 (TGF- β 1)/SMAD pathway. According to the ontology analysis (revised Fig. 7H) of gene containing uTIS with non-AUG as the start codon and activated in the early embryo, some proteins that regulated the mitotic cell cycle process can be BZW1 target gene candidates including *Cdk1*, *Rad21* and *Chek2*. We have further described and discussed the effects of BZW1 on cleavage in the revised manuscript and Figure 8A.

Ref.

Li, S. et al. BZW1, a novel proliferation regulator that promotes growth of salivary mucoepidermoid carcinoma. *Cancer letters* 284, 86-94, doi:10.1016/j.canlet.2009.04.019 (2009).

Chiou, J. et al. Overexpression of BZW1 is an independent poor prognosis marker and its down-regulation suppresses lung adenocarcinoma metastasis. *Scientific reports* 9, 14624, doi:10.1038/s41598-019-50874-x (2019).

Shi, Z., Xiao, C., Lin, T., Wu, J. & Li, K. BZW1 promotes cell proliferation in prostate cancer by regulating TGF-beta1/Smad pathway. *Cell cycle* 20, 894-902, doi:10.1080/15384101.2021.1909242 (2021).

Minor points

1. Fig. 1B: The BZW1 band at 4-cell seems to be at a lower position and associated with a small band.? Or is this true?

Response: We repeated the western blotting with the BZW1 antibody at every stage of the early embryo. The BZW1 expression pattern was the same as before, and two different sizes of BZW1 bands from the zygote to morula stage were detected. Considering that both bands were decreased in *siBzw1* embryos (revised Figure 7C), we hypothesized that BZW1 may have two splicing isoforms or undergo modifications in mouse early embryos.

2. Fig. 3C: This figure needs further explanations. Probably, the mCherry-positive areas of the “21/31” embryo may be important... The authors should explain the points to be focused on in the text or figure legend.

Response: We thank the reviewer for this comment. Considering that the results of the original Figure 3C were not clear, we replaced this panel with new results in the revised Figure 3, which shows that the total E-cad protein level was unchanged after immunofluorescent staining

(revised Figure 3C, D), but the level of E-cad remaining attached to the cytoskeleton (Triton X-100 treated) was lower in the *Bzwl*-knockdown embryos than in the negative control (revised Figure 3C, E).

3. Page 6, line 26: What are the “differentiation markers”? Is this terminology correct?

Response: We apologize for the confusion. We have reworded “differentiation markers” to “lineage-specific genes” in the revised manuscript.

4. Page 7, line 13: Hatching of blastocysts largely depends on the activity of trophoblast. However, the ICM was mostly affected in this case. Please explain this discrepancy.

Response: In our study, some of the *siBzwl* embryos that reached the blastocyst stage contained smaller blastocyst cavities. These morphologically abnormal blastocysts are difficult to hatch out. Other reports have shown that the inner cell mass can direct trophoblast invasion and migration (Ashary, et al, 2017). In some cases, such as *siSmim14* treatment during early embryo development, embryos expressed CDX2 normally but failed to hatch out (Cui, et al, 2016). In addition, *Cdx2* was only one of the trophoblast lineage-specific genes. The trophoblast of *siBzwl* embryos may have defects other than in *Cdx2* expression. We have addressed this issue in the manuscript as suggested by the reviewer.

Ref.

Ashary N, Tiwari A, Modi D. Embryo Implantation: War in Times of Love. *Endocrinology*. 2018 Feb 1;159(2):1188-1198. doi: 10.1210/en.2017-03082.

Cui W, Dai X, Marcho C, Han Z, Zhang K, Tremblay KD, Mager J. Towards Functional Annotation of the Preimplantation Transcriptome: An RNAi Screen in Mammalian Embryos. *Sci Rep*. 2016 Nov 21;6:37396. doi: 10.1038/srep37396.

5. Fig. S2F: What is “ERM” in the Y axis? This figure shows the p-Ezrin intensity.

Response: We apologize for the oversight. This spelling mistake has been corrected in revised Figure S5C.

6. Fig. S2G: I recommend this figure to be associated with Fig. 5G.

Response: We thank the reviewer for the suggestion. We have moved the original Figure S2G into the revised Fig. 6C.

7. Page 9, line 12 “In addition, ...”: Please check the composition of this sentence.

Response: We have rephrased this sentence in the revised manuscript.

Original version: In addition, conventional AUG initiation codons, CUG was the most prominent start codon among these non-AUG in mRNAs that were upregulated at every stage (Fig. S4E).

Revised version: Except for conventional AUG initiation codons, CUG was the most prominent start codon among these non-AUG in mRNAs that were upregulated at every stage (Fig. S6E).

Reviewer #3 (Remarks to the Author):

In their MS entitled: Translation Regulatory Factor BZW1 Regulates Preimplantation Embryo Development and Compaction by Restricting Global non-AUG Initiation, Zhang and colleagues investigate the role of a translational regulator Bzw1 on preimplantation mouse development. Knockdown of this factor in the zygote leads to developmental defects from 8-cell stage, with reduced rates of blastocyst formation, compaction and aberrant morphologies. The authors claim that Bzw1 is necessary for morula compaction and pluripotency factor expression, and also that it regulates global protein synthesis as early as the 2-cell stage through restricting non-AUG translation initiation.

While the developmental phenotype with reduced blastocyst formation seems to be consistent and merits further investigation, the direct experimental evidence in this MS does not support some of the claims laid out by the authors. Importantly, without additional global molecular profiling of embryonic transcription (RNA-seq) or translation, in my view it is impossible to conclude on the origin of the developmental phenotype from the data present in the MS.

Response: We highly appreciate the reviewer's insightful, specific, and constructive comments and suggestions for our manuscript. We sincerely thank the reviewer for pointing out the issues with data description, presentation, and interpretation in our original manuscript. We have made extensive efforts to provide more experimental details, elaborate on our data analyses, interpretations, and conclusions, and improve the quality of the content during revision. We hope that the reviewer finds the revised manuscript suitable for publication.

I outline some of my questions below.

Fig 2: What is the effect of Bzw1 KD on Bzw2 RNA and/or protein levels? Is there functional compensation?

Response: We have provided the changes of *Bzw1* and *Bzw2* mRNA level after *Bzw1* knockdown in the revised Figure 2D. The results showed that *siBzw1* had no significant effect on the *Bzw2* mRNA.

In addition, we added the effect of *siBzw2* on embryonic development. We also showed that BZW1 overexpression partially rescued the developmental defect of *siBzw2* embryos. BZW2 overexpression, in turn, could also rescue the developmental defect of *siBzw1* embryos, suggesting that there are functional compensations between BZW1 and BZW2 to some extent. We have provided these data in revised Figure 2 and S2.

Fig 2C: There seems to be a decrease in developmental rate already between 4 and 8-cell stage of development (significantly lower number of 8-cell stage embryos), which occurs much prior to compaction and cell lineage allocation. Have the authors considered that the global phenotype later in development arises due to an earlier developmental defect, and not compaction?

Response: As the reviewer indicated, the developmental rates in the 4 and 8-cell stages decreased after *Bzw1* KD, which indeed contributed to the global phenotype decline in blastocyst development rate. However, when we calculated the compaction rate, the number of embryos that reached the 8-cell stage was considered to be the cardinal count.

Some studies have reported that BZW1 is highly expressed in cancer tissues and cell lines such as prostate cancer and lung adenocarcinoma. BZW1 promotes cell proliferation in prostate cancer by regulating the transforming growth factor-beta1 (TGF- β 1)/SMAD pathway. According to our ontology analysis (revised Figure 7H) of genes containing uTIS with non-AUG as the start codon and activated in the early embryo, some proteins that regulate the mitotic cell cycle process can be BZW1 target gene candidates including *Cdk1*, *Rad21* and *Chek2*. These evidences suggest that BZW1 may also play an important role in cell proliferation and division.

We have further described and discussed the effects of BZW1 on cleavage in the revised text and revised Figure 8A.

Ref.

Li, S. et al. BZW1, a novel proliferation regulator that promotes growth of salivary mucoepidermoid carcinoma. *Cancer letters* 284, 86-94, doi:10.1016/j.canlet.2009.04.019 (2009).

Chiou, J. et al. Overexpression of BZW1 is an independent poor prognosis marker and its down-regulation suppresses lung adenocarcinoma metastasis. *Scientific reports* 9, 14624, doi:10.1038/s41598-019-50874-x (2019).

Shi, Z., Xiao, C., Lin, T., Wu, J. & Li, K. BZW1 promotes cell proliferation in prostate cancer by regulating TGF-beta1/Smad pathway. *Cell cycle* 20, 894-902, doi:10.1080/15384101.2021.1909242 (2021).

Secondly, culturing unhealthy embryos together with potentially healthy ones will certainly affect the developmental rates of the otherwise perfectly fine embryos.

Response: To avoid this harm indicated by the reviewer, we transferred the embryos into the fresh culture medium every 48 h during embryo culture. We have added this detail in the revised Materials and Methods.

Figure 3: What is the rationale behind injecting 1 of the 2 embryonic blastomeres? The authors write this is to reduce *Bzw1* expression in half of the embryos (I suspect this meant embryo), but they then don't look at biases in cell fate allocation. I am unclear as to the conclusions drawn from these perturbations.

Response: Because the time span of compaction is relatively short and embryo development is difficult to synchronize completely, microinjection of *siBzw1* into one blastomere at the 2-cell stage allowed *Bzw1* knockdown in half of the blastomeres of one embryo during compaction, allowing us to observe the effect of *siBzw1* during compaction in a single embryo.

Indeed, some lineage regulators have been reported to bias cell fate in 2-cell stage such as *LincGET*. However, it has been proven that there is no obvious bias of 2-cell-stage blastomeres

towards the generation of filopodia during compaction at the late 8-cell stage (Fierro-González JC, et al, 2013). In contrast, in our study the microinjection of si*Bzwl* into one blastomere at the 2-cell stage was random. If the biases in cell fate allocation seriously affected the results, the rate of compaction would have a great difference compared to *Bzwl* knockdown in whole embryos. However, this phenotype was not observed. We have made a clearer statement regarding this rationale in the text.

Ref. Fierro-González JC, White MD, Silva JC, Plachta N. Cadherin-dependent filopodia control preimplantation embryo compaction. Nat Cell Biol. 2013 Dec;15(12):1424-33. doi: 10.1038/ncb2875.

Looking at E-Cad levels in siCtrl and si*Bzwl* embryos in Figure 3C, I would have a hard time seeing much difference if the figure wasn't annotated.

Response: We agree with the reviewer that the results in Figure 3C are unclear. Therefore, we replaced this panel with new results in the revised Figure 3, which shows that the total E-cadherin protein level was unchanged by immunofluorescent staining (revised Figure 3C, D), but the level of E-cadherin remaining attached to the cytoskeleton (Triton X-100 treated) was lower in the *Bzwl*-knockdown embryo than in the negative control (revised Fig. 3C, E). To clarify the effect of BZW1 on the polarity of blastomeres, *Bzwl* siRNAs were microinjected into one blastomere of 2-cell stage embryos. The polarity of the blastomeres was observed by pEzrin staining. pEzrin^{T567} was detected at the early 8-cell embryo stage, which mediates the formation of F-actin at the apical surface and promotes embryo compaction and polarization. During apical protein polarization, pEzrin^{T567} concentrated at the center of the cell-contact-free surface to form an apical patch (revised Fig 3E, F-a). However, the pEzrin signal in blastomeres with *Bzwl* knockdown was uniformly distributed on the cell-contact free surface (revised Fig 3E, F-b). The pEzrin signal intensity in half *Bzwl* knockdown embryos was lower than that in control embryos (revised Figure 3G).

Figure 4: Do immunofluorescence panels show individual Z-slices or maximal projection images?

Response: The image, which was acquired on a Zeiss LSM 710 confocal microscope, is the maximum intensity projection of the z-stack. We have added this information to the revised Materials and Methods section.

In any case, there are lower levels of both proteins and RNA, and not only Oct3/4, Nanog and Cdx2 are mis regulated but apparently numerous other genes at an RNA and protein level. Later the authors indicate that P-RNApII levels are fine (which they conclude indicates that transcription is unaffected). What is causing then the downregulation of these indicated genes at the RNA level?

Response: We apologize for the confusion regarding the change in the RNA and protein levels. In original Figure S4D (revised Figure S6C), we showed the transcriptional change by single-

cell RNA-seq in *siBzwl* embryos at the 4-cell stage. The results indicated that 98.5% of the detected genes were almost unchanged (< 2-change fold) after *Bzwl* knockdown. This suggests that the decrease in BZW1 had no significant effect on transcription in the early stages after BZW1 knockdown. However, with embryo development, the degree of translation defects became increasingly serious. The embryos showed abnormal morphology and the expression of some ZGA genes was down-regulated. To clarify this, we added the ZGA gene expression levels in the 4-cell and 8-cell stages respectively tested by RT-PCR in revised Figure 4B-C.

Furthermore, could the authors please clarify Figure 4E?

Response: Thanks for reviewer's suggestion.

We compared the proportion of blastocysts forming ICM colonies or trophoblast cells between the control and *siBzwl* groups. We found that approximately 77% of *Bzwl*-knockdown blastocysts that hatched out failed to form ICM colonies (revised Figure 4F), which was significantly higher than that of the control group. However, the proportions of trophoblast cell adhesion and development were not significant different (revised Figure 4G).

We modified the presentation of the data and made a clearer statement in the revised main text and figure legends.

Figure 5:

Figure 5C shows embryos at different developmental stages (control is a blastocyst and the eIF5 OEx is an 8-cell/morula). Since many eIF5 OEx embryos make it to the blastocyst stage (as per Figure 5A), could the authors show E-Cad levels for developmentally matched embryos?

Response: Thanks for reviewer's suggestion. We added the E-CAD immunofluorescence in revised Figure 5E to show the total E-cadherin and E-cadherin localized in the cytoskeleton in matched embryos.

Fig 5E I am somewhat confused as to why third row panel (rescue experiment) shows markedly fewer embryos in the 72h and 96h timepoints compared to 24h and 48h. Can the authors comment?

Response: During embryos culture in vitro, we transported the embryos into fresh culture medium every 48 h. The embryos in the third-row panel were in large numbers. They were split into two medium drops after 48 h. We have provided DIC images of another medium drop below.

What are the relative endogenous levels of eIF5 during preimplantation development and how to they compare to Bwz1 levels?

Response: We appreciate the reviewer's comment. The translation and transcription levels of eIF5 and BZW1/2 during preimplantation development have been added to the revised Figure 5A-B. The results we are obtained from the published data, GSE70605.

Ref. Zhang C, Wang M, Li Y, Zhang Y. Profiling and functional characterization of maternal mRNA translation during mouse maternal-to-zygotic transition. *Sci Adv.* 2022;8(5):eabj3967. doi:10.1126/sciadv.abj3967

Liu W, Liu X, Wang C, Gao Y et al. Identification of key factors conquering developmental arrest of somatic cell cloned embryos by combining embryo biopsy and single-cell sequencing. *Cell Discov* 2016; 2:16010.

What is the direct evidence that Bzw1 competes with eIF5? The authors could try an in vitro competition assay with a cognate RNA and Bwz1 and eIF5.

Response: Thanks for reviewer's suggestion. We conducted a competition assay to prove that BZW1 competes with eIF5 to interact with PIC. The interaction between BZW1 and the eIF2S subunit of the pre-initiation complex (PIC) was impaired when eIF5 was co-expressed. The results have been added to the revised Figure 6B.

Fig 6: General comment: there is a lot of variability between fluorescence levels in both embryo groups, since this depends on how much mRNA was injected into each embryo, and many other effects (fluorescence bleaching, pH, position of the embryo, etc). Concluding of the efficiency of codon usage based on single plane fluorescent image quantitation of ~30 embryos is not exactly precise molecular data. Furthermore, in figure 6C and D, the authors claim there is a global reduction in translation / protein synthesis upon Bzw1 KD which is certainly not obvious from looking at GFP or mCherry signals in panel 6A. In 6C, it also appears that 2 of the 4 blastomeres of the 4-cell stage embryo there is higher HPG signal. Is this biological or technical? Have the authors attempted to do an OPP incorporation assay?

Response: Thanks for reviewer's suggestion. We conducted the OPP incorporation assay in *Bzw1*-depleted embryos from the 2-cell to 4-cell stage and provided the data in the revised Figure 7F. We agreed that fluorescent image quantitation of embryos is not precise. Therefore,

we used the reporters in which GFP starts with CUG codons and Myc starts with AUG codons, to reflect the biases in initiation codon selection. Reporter expression was detected by western blotting with anti-GFP and anti-Myc antibodies. Approximately 130 embryos were collected in each lane. The results showed that the expression of CTG codon-initiated GFP was up-regulated and that of ATG codon-initiated Myc was down-regulated in *Bzw1* knockdown embryos. The ratio of GFP to Myc increased after *Bzw1*-depletion. These results have been added to the revised Figure 7C

In 6C, authors write: at least 20 embryos were analyzed per stage. In 6D authors show quantitation of 20 control and only 13 siBzw1 embryos. Why is that?

Response: We apologize for the oversight. More than 20 embryos were co-cultured with HPG and stained. However, only 13 embryos in the si*Bzw1* group with normal morphology were included in the quantification of the fluorescent signals. We conducted the OPP incorporation assay of *Bzw1*-depleted embryos from the 2-cell to 8-cell stage and the number of embryos analyzed in every stage is presented in the revised Figure 7G. The results also show that *Bzw1* knockdown embryos had weaker OPP signals than the control group at the 2- to 8-cell stages (revised Figure 7F-G).

REVIEWERS' COMMENTS

Reviewer #1 (Remarks to the Author):

The authors have addressed all the concerns raised by this reviewer.

Reviewer #2 (Remarks to the Author):

The paper has been fully revised with new data, figure rearrangements, and addition of some sentences in Discussion. I have no further comments on this manuscript.

Reviewer #3 (Remarks to the Author):

After revisions, the authors have performed additional experiments and clarified several points in the text and methods which I have requested. Therefore I believe the MS is now scientifically stronger and appropriate for publication in Nature Comms.

I have 2 small points:

- 1) in figure 2F, having an X-axis for siBzw2 and siBzw1+2 without any data there is confusing. Please keep only siCtrl and siBzw1 as that is the data you are showing.
- 2) In a few places the authors write Morular instead of morula (or morulae for plural). Please correct this.

Response to the reviewers' comments

Reviewer #1 (Remarks to the Author):

The authors have addressed all the concerns raised by this reviewer.

Response: Thanks for the reviewer's comments.

Reviewer #2 (Remarks to the Author):

The paper has been fully revised with new data, figure rearrangements, and addition of some sentences in Discussion. I have no further comments on this manuscript.

Response: Thanks for the reviewer's comments.

Reviewer #3 (Remarks to the Author):

After revisions, the authors have performed additional experiments and clarified several points in the text and methods which I have requested. Therefore, I believe the MS is now scientifically stronger and appropriate for publication in Nature Comms.

I have 2 small points:

1) in figure 2F, having an X-axis for siBzw2 and siBzw1+2 without any data there is confusing. Please keep only siCtrl and siBzw1 as that is the data you are showing.

Response: Thanks for the reviewer's comments. The embryos cannot develop to the blastocyst stage in siBzw1 and siBzw2 group. We added the number of counted embryos in figure 2f.

2) In a few places the authors write Morular instead of morula (or morulae for plural). Please correct this.

Response: Thanks for the reviewer's comments. We have corrected the spelling mistakes in all figure.